# Probabilistic Matching of Real and Generated Data Statistics in Generative Adversarial Networks

**Philipp Pilar**                                                                                          *philipp.pilar@it.uu.se*
*Department of Information Technology*
*Uppsala University*

**Niklas Wahlström**                                                                                    *niklas.wahlstrom@it.uu.se*
*Department of Information Technology*
*Uppsala University*

**Reviewed on OpenReview:** *https://openreview.net/forum?id=o1oetBJuSv*

## Abstract

Generative adversarial networks constitute a powerful approach to generative modeling. While generated samples often are indistinguishable from real data, there is no guarantee that they will follow the true data distribution. For scientific applications in particular, it is essential that the true distribution is well captured by the generated distribution. In this work, we propose a method to ensure that the distributions of certain generated data statistics coincide with the respective distributions of the real data. In order to achieve this, we add a new loss term to the generator loss function, which quantifies the difference between these distributions via suitable $f$-divergences. Kernel density estimation is employed to obtain representations of the true distributions, and to estimate the corresponding generated distributions from minibatch values at each iteration. When compared to other methods, our approach has the advantage that the complete shapes of the distributions are taken into account. We evaluate the method on a synthetic dataset and a real-world dataset and demonstrate improved performance of our approach.

## 1 Introduction

Generative adversarial networks (GANs) (Goodfellow et al., 2014) comprise a generator and a discriminator network trained adversarially until the generator manages to produce samples realistic enough to fool the discriminator. Since their conception, GANs have become a popular tool for generative modeling (Hong et al., 2019; Gui et al., 2021). The GAN framework is generally applicable and it is probably best known for its successes in image generation (Reed et al., 2016; Mathieu et al., 2016; Isola et al., 2017; Ledig et al., 2017).

Although GANs have proven powerful, challenges such as mode collapse and non-convergence remain (Saxena and Cao, 2021). It is often the case that the generated samples, while realistic, stem from only a subspace of the true data distribution, or do not reflect the relative frequencies with which they occur accurately. For scientific applications in particular, such as in cosmology (Rodriguez et al., 2018; Villaescusa-Navarro et al., 2021) or high-energy physics (Paganini et al., 2018; Alanazi et al., 2021), where GANs may serve as differentiable surrogate models for expensive but highly accurate numerical simulations, having a good match between the distributions is essential (Kansal et al., 2023).

It is this latter aspect that we tackle in this work, by matching properties of the generated distribution with those of the real data distribution. In particular, we consider statistics of the dataset such as the power spectrum components, and match their distributions. We incorporate these requirements in the form of probabilistic constraints since it is not properties of individual samples that are enforced, but collective characteristics of the dataset. The approach is chiefly aimed at applications in science, where suitable statistics

to be matched can be chosen through domain knowledge. The only requirement on the statistics is that they need to be differentiable.

The main ingredients of our approach are the following: we approximate both the distributions of the real data and the generated data statistics efficiently via kernel density estimation (KDE) (Silverman, 1986). In each iteration, the mismatch between true and generated distributions is then calculated through suitable $f$-divergences and added as an additional term to the generator loss. That way, we end up with a constrained generated distribution. Using $f$-divergences, as opposed to e.g. low-order moments of the distributions, has the advantage that the full shapes of the distributions are taken into account. In the following, we refer to our method as probabilistically constrained GAN (pcGAN).

## 2 Related Work

The field of physics-informed machine learning, where prior knowledge is introduced into the ML model, has been an active area of research in recent years (Karniadakis et al., 2021; Cuomo et al., 2022). In the context of GANs, two main approaches for including prior knowledge in the model exist.

In the first approach, the constrained values can be fed as additional inputs into the discriminator, such that it can explicitly use constraint fulfillment as a means to distinguish between real and generated data. In Stinis et al. (2019), GANs are employed for interpolation and extrapolation of trajectories following known governing equations. The generated trajectories are constrained to fulfill these equations by passing the constraint residuals as additional inputs to the discriminator; in order to prevent the discriminator from becoming too strong, some noise is added to the residuals of the real data, which might otherwise be very close to zero. When extrapolating, the GAN is applied iteratively from some initial condition; in order to train stably, it learns to predict the correct trajectory from slightly incorrect positions of the previous step.

In Yang et al. (2019), a physically-informed GAN (PI-GAN) is developed to model groundwater flow. They make use of the same basic idea as physics-informed neural networks (Raissi et al., 2019) and employ automatic differentiation in order to obtain a partial differential equation (PDE) residual on the GAN output, which is in turn fed into the discriminator. By evaluating the GAN prediction at many different points and comparing to an equivalent ensemble of true values of the corresponding physical field, the GAN is constrained to adhere to a stochastic PDE.

In the second approach, prior knowledge may be taken into account via additional loss terms in either discriminator or generator loss: in Khattak et al. (2018; 2019), GANs are employed to simulate detector signals for high-energy physics particle showers. Here, physical constraints such as the particle energy are taken into account via additional generator loss terms.

In Yang et al. (2021), the incorporation of imprecise deterministic constraints into the GAN is investigated; e.g. the case where the GAN output is supposed to follow a PDE, but where the PDE parameters are not known accurately could be formulated as an imprecise constraint. In a first step, deterministic constraints can be included by adding the constraint residuals as an additional loss term to the generator loss; they argue that it is better to add such terms to the generator since this strengthens the weaker party in the adversarial game, instead of giving an even larger advantage to the discriminator. In order to make the constraint imprecise, they do not require that the residuals go to zero, but instead only include residuals above a certain threshold value $\epsilon^2$ in the loss.

The work closest in aim to ours is probably that by Wu et al. (2020), where a statistical constrained GAN is introduced. They add an additional term to the generator loss function in order to constrain the covariance structure of the generated data to that of the true data. This additional term is a measure of similarity between the covariances, and they concluded that the Frobenius norm was the best choice for this purpose. They use their method to obtain better solutions for PDE-governed systems.

Similar to Wu et al. (2020), our method also imposes probabilistic constraints via an additional term to the generator loss. However, there are significant differences: firstly, our method does not consider the covariance structure of the dataset in particular, but instead allows to constrain on arbitrary statistics of the data. Secondly, our method uses $f$-divergences to match the distributions of true and generated data statistics

explicitly and takes the complete shapes of the distributions into account, instead of only the second-order moments.

## 3 Background

The basic idea of generative adversarial networks (GANs) (Goodfellow et al., 2014) is to train a generator to generate samples of a given distribution and a discriminator (or critic) to distinguish between real and generated data. During the training, both networks are pitted against each other in a minimax game with value function

$$\min_G \max_D V(D, G) = \mathbb{E}_{x \sim p_{\text{data}}(x)} \left[ \log D(x) \right] + \mathbb{E}_{z \sim p_z(z)} \left[ \log(1 - D(G(z))) \right]. \tag{1}$$

Here, $D$ denotes the discriminator, $G$ the generator, $x$ samples drawn from the real data and $z$ randomly generated latent space vectors serving as input to the generator; $p_{\text{data}}$ and $p_z$ denote the real data distribution and the latent vector distribution, respectively. Discriminator and generator are then trained alternatingly (with $m \geq 1$ discriminator updates between each generator update); in (Goodfellow et al., 2014), it is shown that a stable equilibrium to the minimax problem (1) exists and that the optimal solution lies in the generator producing samples from the true data distribution.

The standard GAN can be very difficult to train and often suffers from mode collapse. In Arjovsky et al. (2017), the Wasserstein GAN (WGAN) was introduced, where they suggest the earth-mover (EM) distance as a new loss for the GAN. They show that the discriminator and generator losses can then be expressed as

$$\mathcal{L}_D = D(x_{\text{gen}}) - D(x_{\text{true}}), \tag{2a}$$
$$\mathcal{L}_G = -D(x_{\text{gen}}), \tag{2b}$$

under the condition that the discriminator is Lipschitz continuous. Rather crudely, this is enforced by clipping the weights of the discriminator. In the end, the terms in (2) are approximated as expectations over minibatches.

With this loss function, the discriminator can be interpreted as a critic that assigns scores to both true and generated samples. These scores are not constrained to any specific range and can therefore give meaningful feedback to the generator also when the discriminator is outperforming. Advantages of the WGAN include improved learning stability as well as meaningful loss curves (Gui et al., 2021).

In this work, we also consider two other common variants of the GAN: firstly, the WGAN with gradient penalty (WGAN-GP) (Gulrajani et al., 2017), where the aforementioned weight clipping is avoided by instead imposing a penalty on the discriminator that is supposed to enforce Lipschitz continuity. Secondly, the spectrally normalized GAN (SNGAN) (Miyato et al., 2018), where Lipschitz continuity is ensured by constraining the spectral norm of each layer of the discriminator explicitely.

## 4 Method

The aim of our method is to consider the distributions of $N_s$ differentiable statistics $z$ of the true dataset, such as e.g. components of the power spectrum (compare Appendix C.1), and to ensure that the same statistics, when extracted from the generated data, are distributed equally.

In order to match true ($p_{\text{true}}$) and generated ($p_{\text{gen}}$) distributions, we modify the generator loss (2b) as follows:

$$\mathcal{L}_G^c = \mathcal{L}_G + \lambda \sum_{s=1}^{N_s} \lambda_s h(p_{\text{true}}(z_s), p_{\text{gen}}(z_s)). \tag{3}$$

The function $h$ is an $f$-divergence that quantifies the mismatch between $p_{\text{true}}$ and $p_{\text{gen}}$, $\lambda$ is a global weighting factor for the constraints, and the $\lambda_s$, for which $\sum_s \lambda_s = 1$, allow to weight the constraints individually.

Three important choices remain to be made: how to choose the function $h$, how to obtain suitable functional representations for $p_{\text{true}}$ and $p_{\text{gen}}$, and how to adequately weight the different loss terms.

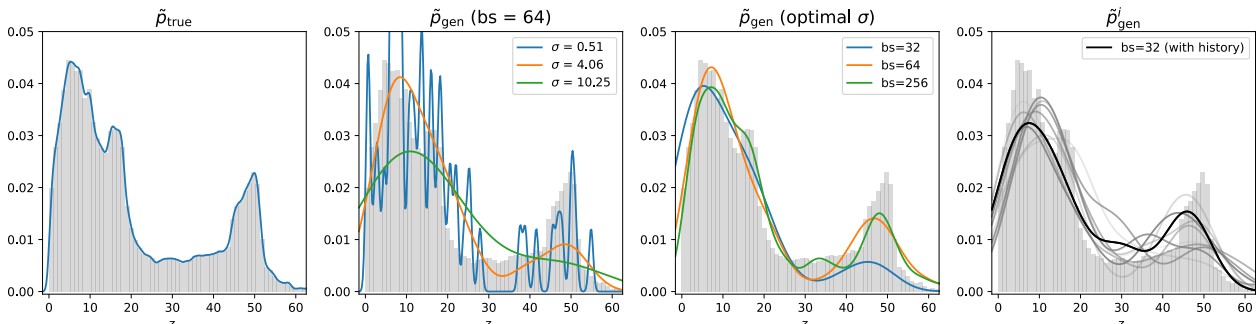

Figure 1: The various representations involved in matching the statistic $z_s$ are depicted. The histogram in the background shows the true data distribution. **Left:** Representation of the true data distribution. **Middle left:** Representation of the generated data distribution with batch size 64 for different choices of $\sigma$ in the kernel. **Middle right:** Representation of the generated data distribution for various batch sizes with optimal choice for $\sigma$ (as determined via Algorithm 3 in the appendix). **Right:** Taking the recent minibatch history into account (here with $\epsilon = 0.9$) can smoothen out fluctuations and lead to a more accurate representation. In this figure, a perfectly trained generator has been assumed, i.e. the minibatches have been sampled from real data.

### 4.1 Quantifying the Mismatch

Let $p$, $q$ be arbitrary probability density functions (PDFs). For $f$-divergences $h$, it holds that $h(p, q) \geq 0$, with equality if and only if $p = q$. These properties justify the use of $f$-divergences for the function $h$ in (3). A major advantage of using $f$-divergences, as opposed to e.g. the Wasserstein distance, is that they are efficient to calculate.

The Kullback-Leibler (KL) divergence constitutes a straightforward choice for $h$ and is defined as

$$h(p, q) = \mathrm{KL}(p||q) = \int_{-\infty}^{\infty} p(x) \log \frac{p(x)}{q(x)} dx. \tag{4}$$

The KL divergence is asymmetric and we consider the forward KL, also known as zero-avoiding, in order to ensure a complete overlap of areas with non-zero probability of the distributions; in case of the reverse, or zero-forcing, KL, the loss term would typically tend to match $q$ to one of the peaks of $p$ and hence fail to match the distributions in a way suitable for our purposes.

The Jeffreys divergence (JD), which can be thought of as a symmetrized version of the KL divergence, as well as the total variation (TV) distance constitute further options:

$$h(p, q) = \mathrm{J}(p||q) = \frac{1}{2} \left( \mathrm{KL}(p||q) + \mathrm{KL}(q||p) \right), \tag{5}$$

$$h(p, q) = V(p - q) = \int_{-\infty}^{\infty} |p(x) - q(x)| dx. \tag{6}$$

An advantage of the latter choice is that no divisions by zero can occur, which may cause problems with the other two options.

As an alternative to using $f$-divergences, we also discuss the maximum mean discrepancy (MMD) as a possible loss function in Appendix B.2. We show that there are drawbacks to using the MMD loss and that the method performs better when using $f$-divergences.

### 4.2 Obtaining Representations

In order to evaluate the loss terms in (3), means of extracting representations for both the true and generated PDFs are required. We denote these representations as $\tilde{p}_{\text{true}}$ and $\tilde{p}_{\text{gen}}$. Note that $\tilde{p}_{\text{true}}$ will need to be determined only once, in advance of the GAN training, since it remains constant. In contrast to the true

distribution, the generated distribution changes during GAN training, and hence $\tilde{p}_{\text{gen}}$ also needs to be determined anew after each generator update.

Kernel density estimation (KDE) (Silverman, 1986) has proven effective for obtaining these representations. For the true distributions, we then get

$$\tilde{p}_{\text{true}}(z_s) = \frac{1}{N} \sum_{j=1}^{N} \frac{1}{\bar{\sigma}_s} K\left(\frac{z_s - z_{sj}}{\bar{\sigma}_s}\right), \qquad (7)$$

where $N$ denotes the number of datapoints, $K$ the kernel function and $\bar{\sigma}_s$ the bandwidth of the kernel. The choice $\bar{\sigma}_{s} = \frac{1}{200}(\max_j(z_{sj}) - \min_j(z_{sj}))$ has proven to give accurate representations for the true distributions (compare e.g. the leftmost plots in Figs. 1, 2, and 5), as we typically have $N \gg 1000$ and can afford to choose such a small value. Throughout the paper, we use Gaussian kernels with $K(x) = (2\pi)^{-1/2}e^{-x^2/2}$. We evaluate different kernel choices in Appendix B.1.

We also approximate the generated distributions at each iteration using KDE, using the constraint values as obtained from the current minibatch samples. That is, we obtain the approximate generated PDFs as

$$\tilde{p}_{\text{gen}}(z_s) = \frac{1}{n_{\text{batch}}} \sum_{j=1}^{n_{\text{batch}}} \frac{1}{\sigma_s} K\left(\frac{z_s - z_{sj}}{\sigma_s}\right), \quad (8)$$

where $n_{\text{batch}}$ denotes the batch size.

For $\tilde{p}_{\text{gen}}(z_s)$, choosing $\sigma_s$ adequately is crucial and requires more thought than in the case of $\tilde{p}_{\text{true}}$. This is due to the fact that there are much fewer samples available in the minibatches. The bandwidths $\sigma_s$ are chosen separately for each constraint $z_s$, under the criterion that $\tilde{p}_{\text{gen}}$ as obtained from minibatches drawn from the true data should have a mismatch as small as possible with $\tilde{p}_{\text{true}}$. Since the optimal values of $\sigma_s$ would be expected to depend both on the range of value $z_s$ in the true dataset and the batch size, we parameterized them as

$$\sigma_s(n_{\text{batch}}) = \text{std}(z_s)/f_\sigma^s(n_{\text{batch}}). \qquad (9)$$

A detailed description of how to determine the optimal values for $f_\sigma^s$ is given in Appendix A.

In order to improve the accuracy of $\tilde{p}_{\text{gen}}$ (assuming that $p_{\text{gen}}$ does not change drastically between subsequent iterations), we can include information from the preceding minibatches via an exponentially decaying historical average:

$$\tilde{p}_{\text{gen}}^i(z_s) = (1 - \epsilon)\tilde{p}_{\text{gen}}(z_s) + \epsilon\tilde{p}_{\text{gen}}^{i-1}(z_s), \qquad (10)$$

where the parameter $\epsilon$ defines the strength of the decay and $i$ denotes the current iteration. In this way, the potentially strong fluctuations between minibatches are smoothened out, allowing for a more accurate representation of $p_{\text{gen}}$.

With representation (10) for the generated distribution and (7) for the true distribution, the one-dimensional integrals required for evaluating $h(\tilde{p}_{\text{true}}, \tilde{p}_{\text{gen}})$ in (3) can be carried out numerically. In Fig. 1, the various representations are illustrated.

---

**Algorithm 1** High-level algorithm

**Step 1:** obtain $\tilde{p}_{\text{true}}$ via (7)
**Step 2:** determine the optimal values $f_\sigma^s$ in (9)
        (see Algorithm 3 in the appendix)
**Step 3:** train the pcGAN (see Algorithm 2)

---

**Algorithm 2** Training the probabilistically constrained GAN (pcGAN)

**Input:** Untrained $D$ and $G$; $\tilde{p}_{\text{true}}$; data $\{x_{\text{true}}\}$; $h$; $\lambda$
**Output:** Trained $D$ and $G$
**for** $i = 1$ **to** $N_{\text{it}}$ **do**
  **for** $k = 1$ **to** $m - 1$ **do**
    sample $x_{\text{true}}$
    generate $x_{\text{gen}}$
    $\mathcal{L}_D = \text{mean}(D(x_{\text{gen}}) - D(x_{\text{true}}))$
    update $D$
    clip weights
  **end for**
  generate $x_{\text{gen}}$
  $\mathcal{L}_G^0 = -\text{mean}(D(x_{\text{gen}}))$
  **for** $s = 1$ **to** $N_s$ **do**
    calculate statistics $\{z_{sj}\}_{j=1}^{n_{\text{batch}}}$ from $x_{\text{gen}}$
    determine $\tilde{p}_{\text{gen}}^i(z_s)$ according to (10)
    $l_s = h(\tilde{p}_{\text{true}}(z_s), \tilde{p}_{\text{gen}}^i(z_s))$
  **end for**
  $\eta = l - \min(l) + 0.1(\max(l) - \min(l))$
  $\lambda_s = \frac{\eta_s}{\sum_{s'} \eta_{s'}}$
  $\mathcal{L}_G^c = \lambda \sum_s \lambda_s l_s$
  $\mathcal{L}_G = \mathcal{L}_G^0 + \mathcal{L}_G^c$
  update $G$
**end for**

---

### 4.3 Weighting the Constraints

The following heuristic scheme has proven effective in weighting the constraints according to how big their mismatches are relative to each other: first, we calculate $\eta = l - \min(l) + 0.1(\max(l) - \min(l))$, where $l$ is a vector with components $l_s = h(\tilde{p}_{\text{true}}(z_s), \tilde{p}^i_{\text{gen}}(z_s))$. Then we assign $\lambda_s = \frac{\eta_s}{\sum_{s'} \eta_{s'}}$. The first term in the definition of $\eta$ quantifies the constraint fulfillment relative to the best-fulfilled one and the second term prevents already well-matched constraints from ceasing to be included. The global weighting factor $\lambda$ needs to be tuned separately.

A high-level overview of the method is given in Algorithm 1 and the pcGAN training is detailed in Algorithm 2. Note that, while our training algorithm is based on the WGAN, the modified generator loss (3) is more general and can be used for other types of GANs as well.

## 5 Results [1]

In this section, we present the results obtained with our model. We introduce a set of evaluation metrics and consider a synthetic example and a real-world dataset from physics. The evaluation metrics are chosen to cover different aspects of the generated distribution and evaluate the GAN performance both in the sample space and in a lower-dimensional space of high-level features, as is common practice (Kansal et al., 2023). We compare the pcGAN to the unconstrained WGAN, WGAN-GP, SNGAN, and the statistical constrained GAN from Wu et al. (2020). We investigate the impact that training parameters have on the model performance and we combine the probabilistic constraint with the different GAN variants to evaluate its potential for improving their performance.

More results are given in the appendices. In Appendix B.1, we investigate the impact that the choice of kernel for the KDE has on the model performance. In Appendix B.2, we evaluate how well the MMD loss would perform instead of the $f$-divergences for matching the constraints. A discussion on the training time required for the different models is given in B.4. Additional information on the datasets, the training parameters, and the high-level features is given in Appendix C.

### 5.1 Evaluation Metrics

To compare the different models, we consider four evaluation metrics:

The Fréchet distance in the sample space, as an alternative to the widely used Fréchet Inception distance (Heusel et al., 2017). It quantifies the agreement of the first and second-order moments of the real and generated distribution and is calculated via

$$d_F^2 = \|\mu - \mu'\|^2 + \text{Tr}\left(\Sigma + \Sigma' - 2\sqrt{\Sigma\Sigma'}\right), \tag{11}$$

where $\mu$, $\Sigma$ correspond to the true distribution and $\mu'$, $\Sigma'$ to the generated distribution.

The F1-score in the space of high-level features, which is defined as the harmonic mean between precision (P) and recall (R):

$$F_1 = 2\frac{PR}{P + R}. \tag{12}$$

In the context of generative modeling, the precision is the fraction of generated samples that lie in the real data manifold and the recall gives the fraction of real data samples that lie in the generated data manifold Sajjadi et al. (2018). They are calculated as suggested in Kynkäänniemi et al. (2019), with choice $k = 10$ for the k-nearest neighbor.

The agreement of the distributions of the constrained statistics, by calculating the average of the total variations of the differences between their histograms:

$$\bar{V}_c = \frac{1}{N_s}\sum_{s=1}^{N_s} V(p^{\text{hist}}_{\text{true}}(z_s) - p^{\text{hist}}_{\text{gen}}(z_s)), \tag{13}$$

---

[1]The code for the project is available on GitHub: https://github.com/ppilar/pcGAN.

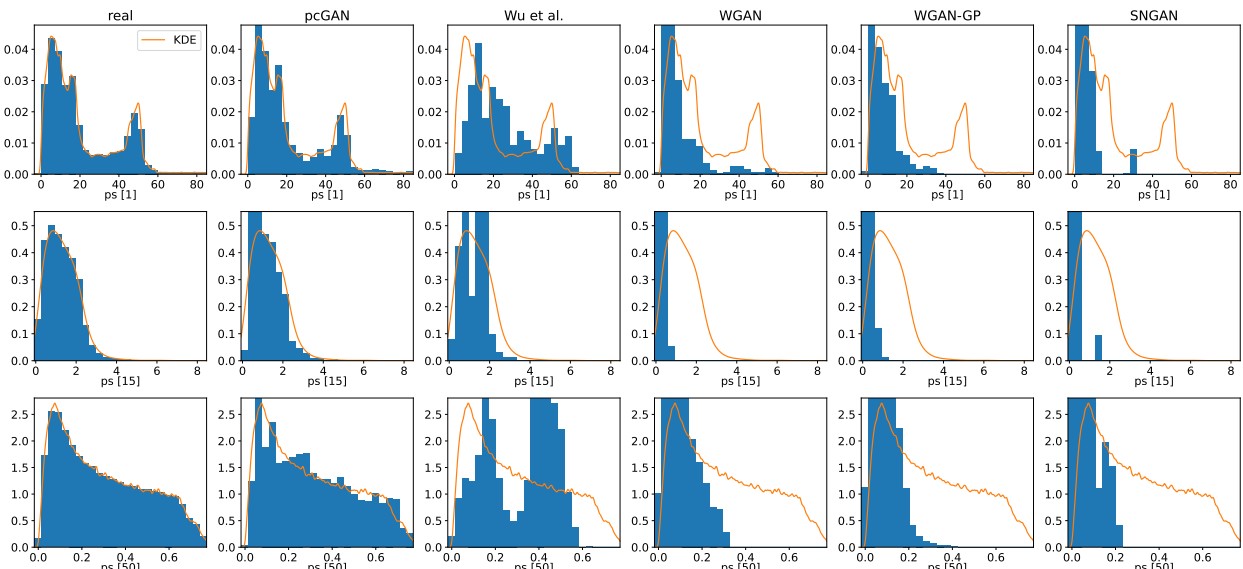

Figure 2: **(Synthetic example)** The distributions of three different power spectrum components ps as obtained by the different models are depicted, where the orange lines show the true distribution as obtained via KDE (7). From left to right, the histograms correspond to the real data, the pcGAN, the method of Wu et al. (2020), WGAN, WGAN-GP, and SNGAN. For the histograms, 20 000 generated samples have been considered (or the full dataset, in case of the real distribution). Parameters for the pcGAN: bs = 256, $\lambda = 500$, $\epsilon = 0.9$, $h = \text{KL}$.

where $p_{\text{true}}^{\text{hist}}$ and $p_{\text{gen}}^{\text{hist}}$ are given by the outline of the histograms in e.g. Fig. 2. Here, the histograms have been chosen instead of KDE, in order to use a quantity that is not directly constrained (and hence without the danger of overfitting to).

The agreement between the distributions of the $N_f$ high-level features (here denoted as $x$). We proceed in the same way as for the constrained statistics:

$$\bar{V}_f = \frac{1}{N_f} \sum_{f=1}^{N_f} V(p_{\text{true}}^{\text{hist}}(x_f) - p_{\text{gen}}^{\text{hist}}(x_f)). \tag{14}$$

To get an idea of the complexity of each metric, it helps to consider them in the following way: the F1 score takes the full shape of the data distribution into account, $d_F$ the first two moments, and $\bar{V}_c$ and $\bar{V}_f$ the marginal distributions of the constrained statistics and the chosen set of high-level features, respectively.

## 5.2 Synthetic Example

For our first experiment, we consider a superposition of sine waves. Each wave consists of two sine waves, $x = \frac{1}{2} \sum_{i=1}^{2} \sin(\omega_i t)$, with angular frequencies sampled randomly from $\omega_i \sim |\mathcal{N}(1,1)|$, and we generate 200 equally-spaced measurements in the interval $t \in [0, 20]$. In total, we create 100 000 samples of size 200 to serve as training data. We perform the Fourier transform for real-valued inputs for each time series in the dataset and we use the square roots of the power spectrum components (i.e. the absolute values of the Fourier coefficients) as the statistics to constrain when training the GAN; that is, we have 101 separate constraints (compare Appendix C.1).

In Figure 2, results for the different GAN variants are depicted. The data generated by the pcGAN matches the true distributions very well. The method of Wu et al. (2020) comes in second, managing to cover the correct range of constraint values, but failing to adhere to the precise shapes of the PDFs. The unconstrained

| | WGAN | WGAN + pc (pcGAN) | Wu et al. | Wu et al. + pc | WGAN-GP | WGAN-GP + pc | SNGAN | SNGAN + pc |
|---|---|---|---|---|---|---|---|---|
| $\frac{d_F^2}{100}$ ($\downarrow$) | 48.94±6.65 | 12.32±2.99 | 3.81±2.13 | **2.49**±0.75 | 49.38±5.26 | 3.50±0.76 | 49.67±5.76 | 12.67±3.51 |
| $F_1$ ($\uparrow$) | 0.15±0.06 | 0.18±0.05 | 0.16±0.03 | **0.20**±0.06 | 0.16±0.04 | 0.19±0.05 | 0.16±0.09 | 0.18±0.08 |
| $\bar{V}_c$ ($\downarrow$) | 1.11±0.10 | 0.20±0.02 | 0.47±0.22 | **0.17**±0.03 | 1.07±0.09 | 0.20±0.01 | 1.40±0.12 | 0.34±0.04 |
| $\bar{V}_f$ ($\downarrow$) | 1.70±0.10 | 0.80±0.05 | 0.95±0.12 | 0.75±0.05 | 1.61±0.15 | **0.70**±0.05 | 1.63±0.11 | 0.88±0.08 |

Table 1: **(Synthetic example)** The different GAN variants and their combinations with the probabilistic constraint are evaluated via different performance metrics, defined in Section 5.1: the Fréchet distance $d_F$, the F1 score, the agreement of the constraint distributions $\bar{V}_c$, and the agreement of the distributions of a selection of high-level features $\bar{V}_f$. The arrows indicate whether high or low values are better. Ten runs have been conducted per model, and the mean values plus-or-minus one standard deviation are given. Bold font highlights best performance. Parameters: bs = 256, $\epsilon = 0.9$, $h = $ KL, and $\lambda = [500, 500, 500, 2500]$, respectively, from left to right for the constrained variants.

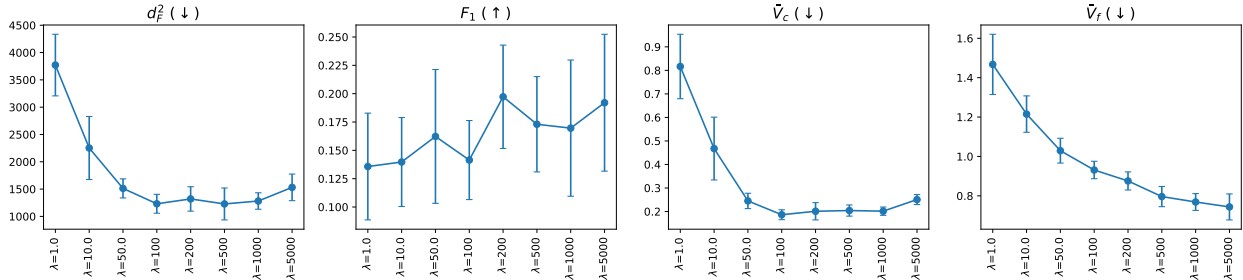

Figure 3: **(Synthetic example)** Different values of the weighting coefficient $\lambda$ are considered (with bs = 256, $\epsilon = 0.9$, $h = $ KL). Ten runs have been conducted per model, and the mean values plus-or-minus one standard deviation are depicted.

WGAN, WGAN-GP and SNGAN are distinctly worse and tend to assign too much weight to the highest peak of the distribution.

In Table 1, we evaluate the performance of the different models in terms of the evaluation metrics defined in Section 5.1. The results for the constraint distributions are well reflected here and the unconstrained versions of the GAN tend to perform worse than both the pcGAN and the method of Wu et al. (2020) on metrics other than the F1 score. When considering the F1 score, pcGAN is ahead of the unconstrained models. Between the pcGAN and Wu et al. (2020), pcGAN outperforms Wu et al. (2020) in all metrics apart from $d_F$, where the method of Wu et al. (2020) is slightly better. This makes sense since $d_F$ only evaluates agreement of the first and second-order moments of the distribution; the latter are precisely what the method of Wu et al. (2020) constrains.

In addition to the pcGAN, results for combinations of the other GAN variants with the probabilistic constraint are also given in the table. It is apparent that adding the constraint also leads to improved performance of the other models. In Appendix B.3, a plot visualizing the constraint fulfillment equivalent to Fig. 2 is given for the different constrained GANs.

In Fig. 3, the impact of the global weighting factor $\lambda$ is investigated. A clear improvement with increasing $\lambda$ is visible for most of the metrics, up to $\lambda \approx 100$. Overall, the value $\lambda = 500$ appears to be a good choice. The fact that $d_F$ and $F_1$ also improve indicates that the constraints help to produce a better diversity of samples.

In Fig. 4, we consider the impact that the batch size and the historical averaging have on the results. Both $d_F$ and constraint fulfillment improve with increasing batch size, although we observe diminishing returns for batch sizes larger than 256. The inclusion of historical averaging improves $d_F$, with higher values of $\epsilon$ yielding larger improvements, whereas constraint fulfillment $\bar{V}_c$ is only weakly affected by the choice of $\epsilon$ and the

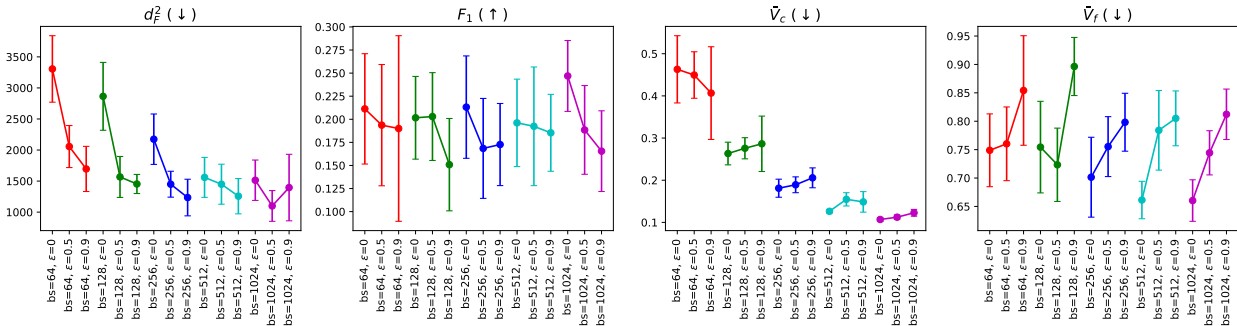

Figure 4: **(Synthetic example)** Different batch sizes with and without historical averaging are considered (with $\lambda = 500$, $h = \mathrm{KL}$). The different colors indicate which points belong to the same batch size. Ten runs have been conducted per model, and the mean values plus-or-minus one standard deviation are depicted.

| | h=TV, $\epsilon$=0.5 | h=TV, $\epsilon$=0.9 | h=KL, $\epsilon$=0.5 | h=KL, $\epsilon$=0.9 | h=JD, $\epsilon$=0.5 | h=JD, $\epsilon$=0.9 |
|---|---|---|---|---|---|---|
| $\frac{d_F^2}{100}$ ($\downarrow$) | 0.21±0.06 | 0.15±0.03 | 0.15±0.02 | **0.12**±0.02 | 0.15±0.05 | 0.14±0.04 |
| $F_1$ ($\uparrow$) | 0.15±0.04 | 0.13±0.03 | 0.17±0.05 | 0.18±0.04 | **0.18**±0.07 | 0.16±0.06 |
| $\bar{V}_c$ ($\downarrow$) | 0.29±0.05 | 0.25±0.06 | 0.19±0.02 | 0.19±0.02 | **0.14**±0.01 | 0.16±0.02 |
| $\bar{V}_f$ ($\downarrow$) | 0.81±0.07 | 0.80±0.08 | 0.76±0.05 | 0.79±0.06 | **0.75**±0.07 | 0.86±0.08 |

Table 2: **(Synthetic example)** Different choices for the $f$-divergence $h$ quantifying the mismatch between $p_{\text{true}}$ and $p_{\text{gen}}$ in (3) are considered (with bs = 256, $\lambda = 500$, $h = \mathrm{KL}$). Ten runs have been conducted per model, and the mean values plus-or-minus one standard deviation are given. Bold font highlights best performance.

metric $\bar{V}_f$ is negatively affected. $F_1$ seems to be largely unaffected by the batch size and somewhat negatively affected by large values of $\epsilon$. The larger the batch size, the smaller the impact of historical averaging.

In Table 2, the different options for the $f$-divergence $h$ used for matching the statistics are evaluated. The results indicate that the Jeffreys divergence performs slightly better than the KL divergence, and the total variation is notably worse than the other two options. Furthermore, we observe that increasing the factor $\epsilon$ for the historical averaging tends to improve the results for the total variation and the KL divergence, but slightly decreases the performance in case of Jeffreys divergence.

We conclude that the probabilistic constraint holds promise for improving the performance of many different GAN variants. When training the pcGAN, larger batch sizes are advantageous. For smaller batch sizes, the historical averaging can yield improvements. When choosing the $f$-divergence for matching the constraints, the KL divergence or the Jeffreys divergence should be selected rather than the total variation. The weighting parameter $\lambda$ is essential to consider when tuning the pcGAN.

The architectures used for the discriminator and generator were inspired by the DCGAN architecture and the ADAM optimizer (Kingma and Ba, 2015) was used for optimization. A discussion on the runtime of the different models is given in Appendix B.4. A detailed description of the architecture, settings for the training procedure, and samples as obtained from the different models can be found in Appendix C.2.

### 5.3 IceCube-Gen2 Radio Signals

The IceCube neutrino observatory (Aartsen et al., 2017) and its planned successor IceCube-Gen2 (Aartsen et al., 2021) are located at the South Pole and make use of the huge ice masses present there in order to detect astrophysical high-energy neutrinos. Deep learning methodology has already been employed to extract information such as shower energy or neutrino direction from radio-detector signals (Glaser et al., 2023; Holmberg, 2022). Holmberg (2022) also investigated the use of GANs to simulate detector signals. We are going to consider the filtered Askaryan radio signals from Holmberg (2022), which were generated using

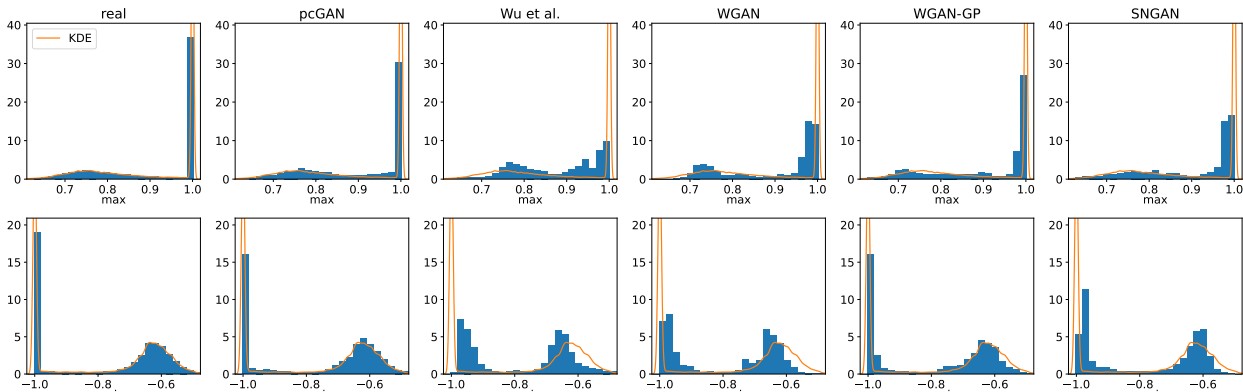

Figure 5: **(IceCube-Gen2)** The distributions of minimum and maximum values as obtained by different models are compared, where the orange lines show the true distribution as obtained via KDE (7). From left to right, the histograms correspond to the real data, the pcGAN, the method of Wu et al. (2020), WGAN, WGAN-GP, and SNGAN. For the histograms, 20 000 generated samples have been considered (or the full dataset, in case of the real distribution). Parameters for the pcGAN: bs = 256, $\lambda = 2$, $\epsilon = 0.9$, $h = $ KL.

| | WGAN | WGAN + pc (pcGAN) | Wu et al. | Wu et al. + pc | WGAN-GP | WGAN-GP + pc | SNGAN | SNGAN + pc |
|---|---|---|---|---|---|---|---|---|
| $d_F^2$ ($\downarrow$) | 25.34±12.85 | 16.71±9.52 | 15.82±8.15 | 14.21±6.20 | **5.36**±6.09 | 6.88±4.80 | 9.02±7.07 | 7.09±3.17 |
| $F_1$ ($\uparrow$) | 0.35±0.17 | 0.38±0.12 | 0.36±0.08 | 0.36±0.06 | **0.48**±0.08 | 0.45±0.06 | 0.46±0.04 | 0.46±0.06 |
| $\bar{V}_c$ ($\downarrow$) | 0.97±0.11 | 0.30±0.06 | 0.99±0.12 | 0.30±0.05 | 0.71±0.12 | **0.16**±0.04 | 1.01±0.06 | 0.17±0.04 |
| $\bar{V}_f$ ($\downarrow$) | 0.76±0.09 | 0.70±0.08 | 0.75±0.05 | 0.68±0.07 | 0.66±0.06 | 0.65±0.07 | 0.65±0.05 | **0.63**±0.04 |

Table 3: **(IceCube-Gen2)** The different GAN variants and their combinations with the probabilistic constraint are evaluated via different performance metrics, defined in Section 5.1: the Fréchet distance $d_F$, the F1 score, the agreement of the constraint distributions $\bar{V}_c$, and the agreement of the distributions of a selection of high-level features $\bar{V}_f$. The arrows indicate whether high or low values are better. Ten runs have been conducted per model, and the mean values plus-or-minus one standard deviation are given. Bold font highlights best performance. Parameters: bs = 256, $\lambda = 2$, $\epsilon = 0.9$, $h = $ KL.

the NuRadioMC code (Glaser et al., 2020) according to the ARZ algorithm (Alvarez-Muñiz et al., 2010). These signals take the form of 1D waveforms and in our experiments we want to focus solely on the shape of these waves, not their absolute amplitudes; this is achieved by normalizing each signal to its maximum absolute value. We use the pcGAN to constrain the generated data on the distributions of the minimum and maximum values of the signals.

The results are depicted in Fig. 5. The pcGAN matches the characteristics of both minimum and maximum distribution well. In particular, it manages to match the spikes at -1 and 1 more accurately than any of the other models. The distributions as obtained via the other models also exhibit the two peaks in each of the distributions but do not reproduce their precise shapes correctly. Out of the remaining models, WGAN-GP matches the distributions best, with only slightly less pronounced spikes at -1 and 1 than the pcGAN. A plot showing the constraint fulfillment for the different constrained GANs is given in Appendix B.3.

In Table 3, the evaluation metrics are given for the different GAN variants together with their constrained versions. The constraints are matched well for all of the constrained models. In terms of the remaining metrics, adding the constraint yields improvements for WGAN, Wu et al. and SNGAN. For WGAN-GP, on the other hand, a slight decrease in performance can be observed. While WGAN-GP performs best on $d_F^2$ and $F_1$, WGAN-GP + pc is the best choice for overall performance when taking constraint fulfillment into account.

The network architecture used for the GANs is based on that from Holmberg (2022). More details on the training procedure, as well as plots of generated samples, are given in Appendix C.3.

## 6 Conclusions and Future Work

We have presented the probabilistically constrained GAN (pcGAN), a method to incorporate probabilistic constraints into GANs. The method is expected to be particularly useful for scientific applications, where it is especially important for generated samples to represent the true distribution accurately and where suitable statistics to be matched can be identified through domain knowledge. For a given statistic $z$, this is achieved by adding the mismatch between the corresponding true and generated distribution as quantified via a suitable $f$-divergence to the generator loss. Kernel density estimation is employed to obtain representations for the distributions of the statistic. By adequately weighting the different loss terms, a large number of statistics can be matched simultaneously.

We have evaluated our method using two different datasets. Our experiments clearly demonstrate that the probabilistic constraint is effective at matching the chosen dataset statistics. In terms of the evaluation metrics, the pcGAN constitutes a significant improvement over the standard WGAN. Depending on the dataset under consideration, it can also outperform WGAN-GP, SNGAN, and the method of (Wu et al., 2020). Combining the probabilistic constraint with GAN variants other than the WGAN also improves the respective models in most cases.

For future work, it would be interesting to extend the method to consider the joint distribution of different statistics, in order to also include correlations between them in the constraint. Furthermore, it would be important to find a way to make the method compatible with conditional GANs in order to widen the range of possible applications. Finding automated ways for obtaining suitable statistics to match, e.g. by using features of classifier networks, could improve the approach and would allow for its application to situations where insufficient domain knowledge is available. In principle, the probabilistic constraint could also be added to other types of generative models. The main requirements would be that new samples are generated during each iteration of the training procedure and that a suitable spot for adding the loss term can be identified. Investigating the applicability of the approach to other generative models, such as autoencoders (Kingma and Welling, 2014) or denoising diffusion probabilistic models (Ho et al., 2020), therefore constitutes another promising avenue for future research.

## Acknowledgements

The work is financially supported by the Swedish Research Council (VR) via the project *Physics-informed machine learning* (registration number: 2021-04321). We thank the anonymous reviewers from TMLR for their constructive feedback, and Jens Sjölund for comments on an early manuscript. Our special thanks also go to Christian Glaser, for interesting discussions and for providing access to the IceCube data generator.

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

## A  Algorithm to Determine $f_\sigma^s$

We propose a method to determine optimal values for $f_\sigma^s$ that is based on the assumption that data sampled from the true distribution should on average have the best possible match between the true distribution and the KDE approximation obtained via the minibatches.

The procedure is summarized in Algorithm 3. In order to determine the optimal value of $f_\sigma^s$ for a given constraint $z_s$ in (9), we perform a grid search over possible values $f_\sigma$. Introducing the standard deviation of $z_s$ into the definition of $\sigma$ via $c$ helps to narrow the range in which the optimal values $f_\sigma^s$ lie. The grid is defined in the array $a_{f_\sigma}$. For each value of $f_\sigma$, we evaluate the mismatch between the true distribution $\tilde{p}_{\text{true}}(z_s)$ (7) and the generated distribution $\tilde{p}_{\text{gen}}(z_s)$ (8) via the $f$-divergence $h$.

---

**Algorithm 3** Determining the optimal value of $f_\sigma^s$

**Input:** true data $\{z_s\}$; $\tilde{p}_{\text{true}}(z_s)$, $h$, $N_{\text{avg}}$
**Output:** $f_\sigma^s$
$c = \text{std}(z_s)$
$a_{f_\sigma} = \text{logspace}(-1, 2, 200)$
**for** $i_N \in [0, N_{\text{avg}}); i_{f_\sigma} \in [0, \text{len}(a_{f_\sigma})]$; **do**
    sample minibatch $\{z_s\}$
    $\sigma = \frac{c}{a_{f_\sigma}[i_{f_\sigma}]}$
    determine $\tilde{p}_{\text{gen}}(z_s)$ according to (8) using $\sigma$
    $\text{H}[i_{f_\sigma}, i_N] = h(\tilde{p}_{\text{true}}(z_s), \tilde{p}_{\text{gen}}(z_s))$
**end for**
$i_{f_\sigma} = \min\left(\text{mean}\left(\text{H}, \dim = 1\right)\right)$
$f_\sigma^s = a_{f_\sigma}[i_{f_\sigma}]$

---

The minibatches are sampled from the true data since the aim is to obtain a mismatch as small as possible for true data. The obtained values for the mismatch are then averaged over $N_{\text{avg}}$ minibatches. Subsequently, the value $f_\sigma$ corresponding to the minimum mean value is determined; this value is the desired optimal value $f_\sigma^s$. Figures 8 and 9 illustrate the grid search.

## B  Additional Results

In this appendix, we present additional experiments and results.

### B.1  The Choice of Kernel

Here, we investigate the impact that the specific choice of kernel for approximating the distributions in (3) has on the performance of the pcGAN. We consider the following kernels (compare Table 4): Gaussian (G), uniform (u), Epanechnikov (epa), triweight (tri), and cosine (cos). The kernel bandwidths are obtained via (9) and Algorithm 3. The results are depicted in Table 5.

The choice of kernel does not have a big impact on the model performance. Overall, the Gaussian kernel seems to be the best choice as it consistently performs well for all of the metrics. A potential reason why the Gaussian kernel is superior can be found in its unbounded support. This means that there will always be some overlap between real and generated distributions, as obtained via KDE, enabling more informative gradients.

| kernel | | |
|---|---|---|
| G | $K(x) = \frac{1}{\sqrt{2\pi}} e^{-\frac{x^2}{2}}$ | |
| u | $K(x) = \begin{cases} \frac{1}{2} \\ 0 \end{cases}$ | $\begin{matrix} \lvert x \rvert \leq 1 \\ \lvert x \rvert > 1 \end{matrix}$ |
| epa | $K(x) = \begin{cases} \frac{3}{4}(1 - x^2) \\ 0 \end{cases}$ | $\begin{matrix} \lvert x \rvert \leq 1 \\ \lvert x \rvert > 1 \end{matrix}$ |
| tri | $K(x) = \begin{cases} \frac{35}{32}(1 - x^2)^3 \\ 0 \end{cases}$ | $\begin{matrix} \lvert x \rvert \leq 1 \\ \lvert x \rvert > 1 \end{matrix}$ |
| cos | $K(x) = \begin{cases} \frac{\pi}{4}\cos\left(\frac{\pi}{2}x\right) \\ 0 \end{cases}$ | $\begin{matrix} \lvert x \rvert \leq 1 \\ \lvert x \rvert > 1 \end{matrix}$ |

Table 4: Different choices of kernel.

### B.2  Using Maximum Mean Discrepancy to Match Statistics

The maximum mean discrepancy (MMD) is a kernel-based statistical test that can be employed to determine whether two distributions are the same (Gretton et al., 2012). It has been used as a loss function to establish generative moment matching networks, a distinct class of generative models (Li et al., 2015; Dziugaite et al., 2015). While an adversarial approach has been suggested to improve MMD networks by learning more suitable kernels (Li et al., 2017), they constitute their own model class and not an extension of the GAN. In

|  | $K = \mathrm{G}$ | $K = \mathrm{u}$ | $K = \mathrm{epa}$ | $K = \mathrm{tri}$ | $K = \cos$ |
|---|---|---|---|---|---|
| $\frac{d_F^2}{100}$ ($\downarrow$) | 13.39±3.08 | 19.73±3.57 | 15.33±4.60 | 15.15±5.70 | **11.69**±1.83 |
| $F_1$ ($\uparrow$) | 0.17±0.06 | 0.13±0.04 | **0.20**±0.05 | 0.14±0.03 | 0.18±0.04 |
| $\bar{V}_c$ ($\downarrow$) | **0.20**±0.02 | 0.21±0.02 | 0.31±0.08 | 0.25±0.06 | 0.23±0.03 |
| $\bar{V}_f$ ($\downarrow$) | **0.80**±0.07 | 0.89±0.06 | 0.84±0.04 | 0.84±0.05 | 0.82±0.04 |

Table 5: **(Synthetic example)** The pcGAN with different choices of kernel $K$ is evaluated via different performance metrics, defined in Section 5.1: the Fréchet distance $d_F$, the F1 score, the agreement of the constraint distributions $\bar{V}_c$, and the agreement of the distributions of a selection of high-level features $\bar{V}_f$. The arrows indicate whether high or low values are better. Ten runs have been conducted per model, and the mean values plus-or-minus one standard deviation are given. Bold font highlights best performance. Parameters: bs = 256, $\lambda = 500$, $\epsilon = 0.9$, $h = \mathrm{KL}$.

|  | $\sigma_0 = 1$ 
 $\lambda = 0.1$ | $\sigma_0 = 1$ 
 $\lambda = 0.5$ | $\sigma_0 = 1$ 
 $\lambda = 1.0$ | $\sigma_0 = 1$ 
 $\lambda = 5.0$ | $\sigma_0 = 0.5$ 
 $\lambda = 0.5$ | $\sigma_0 = 2$ 
 $\lambda = 0.5$ | $\sigma_0 = 5$ 
 $\lambda = 0.5$ | $\sigma_0 = [1, 2, 5]$ 
 $\lambda = 0.5$ |
|---|---|---|---|---|---|---|---|---|
| $\frac{d_F^2}{100}$ ($\downarrow$) | 21.52±3.97 | 18.42±3.03 | 18.59±3.47 | 23.26±2.93 | 19.52±2.12 | **17.27**±4.13 | 18.16±5.63 | 17.63±3.47 |
| $F_1$ ($\uparrow$) | 0.14±0.05 | 0.11±0.04 | 0.11±0.03 | 0.10±0.07 | 0.13±0.04 | 0.17±0.06 | **0.18**±0.04 | 0.13±0.04 |
| $\bar{V}_c$ ($\downarrow$) | 0.56±0.05 | 0.71±0.09 | 0.76±0.10 | 0.96±0.06 | **0.50**±0.03 | 0.90±0.12 | 1.05±0.15 | 0.77±0.05 |
| $\bar{V}_f$ ($\downarrow$) | 1.20±0.11 | 1.13±0.06 | 1.07±0.07 | **1.01**±0.06 | 1.08±0.06 | 1.10±0.08 | 1.10±0.13 | 1.07±0.05 |

Table 6: **(Synthetic example)** The pcGAN with MMD loss is evaluated for different weighting factors $\lambda$ and kernel widths $\sigma_0$ via different performance metrics, defined in Section 5.1: the Fréchet distance $d_F$, the F1 score, the agreement of the constraint distributions $\bar{V}_c$, and the agreement of the distributions of a selection of high-level features $\bar{V}_f$. The arrows indicate whether high or low values are better. Ten runs have been conducted per model, and the mean values plus-or-minus one standard deviation are given. Bold font highlights best performance.

this appendix, we do not consider MMD networks but explore instead the effectiveness of using MMD as the loss function for matching the high-level statistics. That is, we use the MMD loss instead of $f$-divergences (compare Section 4.1) in (3).

The kernel maximum mean discrepancy between two distributions is defined as

$$\mathrm{MMD}^2 = \frac{1}{\sigma}\mathbb{E}\left[K\left(\frac{X - X'}{\sigma}\right) - 2K\left(\frac{X - Y}{\sigma}\right) + K\left(\frac{Y - Y'}{\sigma}\right)\right], \tag{15}$$

where $X$ denotes real data and $Y$ generated data. This leads to the following loss function, where we estimate the expectations over minibatches and omit constant terms (i.e. terms that do not contain $Y$):

$$\mathcal{L}_{\mathrm{MMD}} = \frac{1}{M(M - 1)\sigma}\sum_{m \neq m'} K\left(\frac{y_m - y_{m'}}{\sigma}\right) - \frac{2}{MN\sigma}\sum_{m=1}^{M}\sum_{n=1}^{N} K\left(\frac{y_m - x_n}{\sigma}\right), \tag{16}$$

where $M$ is the number of generated samples and $N$ the number of real samples in the current minibatches.

One drawback of this approach is the mixed loss term in equation (16); it would be too computationally costly to take into account the entire dataset at each iteration wherefore we also need to batch the real data. When using $f$-divergences in loss (3) of our approach, similar problems can be circumvented by evaluating $p_{\mathrm{true}}$ once on a fixed grid in advance of the training. Here, the same trick does not work, since the statistics as extracted from the generated data determine the points at which the kernel $K$ needs to be evaluated.

The results for the MMD loss are given in Table 6. We consider different weighting factors for the loss term, as well as Gaussian kernels with different bandwidth. The bandwidths of the Gaussian kernels for the different constraints are given by the values $\sigma_s$ in (9) times the factors $\sigma_0$ given in the figure. When multiple factors are given, then the sum of the corresponding Gaussian kernels is used. Both in terms of matching the constraints and in terms of the performance metrics, the method performs better than the standard WGAN, but worse than the pcGAN (compare Table. 1).

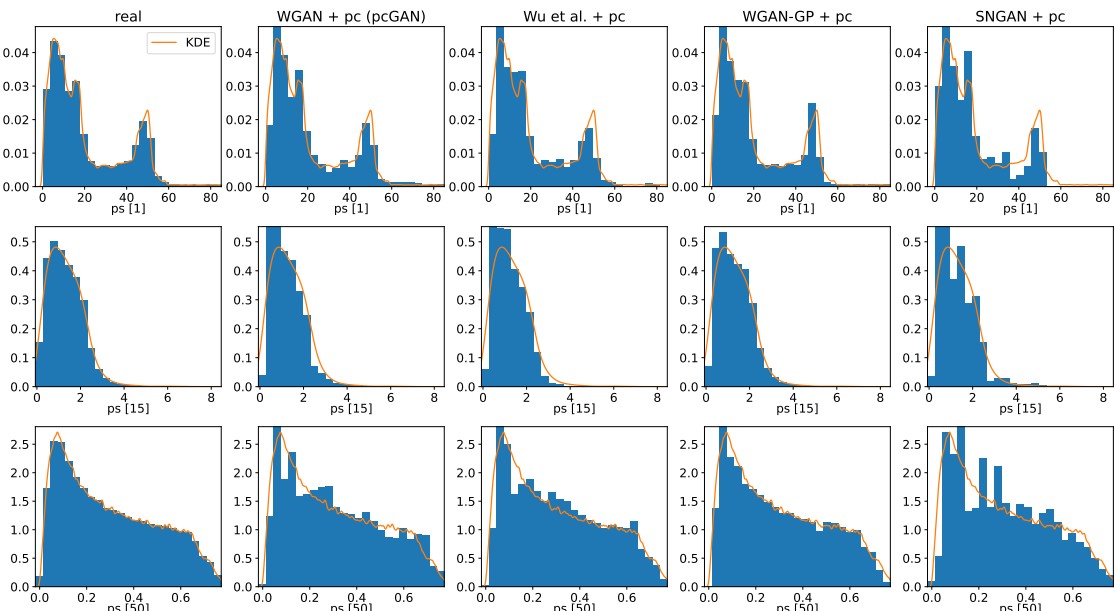

Figure 6: **(Synthetic example)** The distributions of three different power spectrum components ps as obtained with the different GAN variants when combined with the probabilistic constraint are depicted, where the orange lines show the true distribution as obtained via KDE (7). For the histograms, 20 000 generated samples have been considered (or the full dataset, in case of the real distribution). Parameters: bs = 256, $\epsilon = 0.9$, $h$=KL, and $\lambda = [500, 500, 500, 2500]$, respectively, from left to right.

### B.3 Additional plots

In Fig. 6, plots of the constraint fulfillment for the different GAN variants combined with the probabilistic constraint are given. It is apparent that all the constrained models reproduce the constraint distributions well, with WGAN-GP + pc giving the smoothest fit. The distributions obtained with SNGAN + pc are a bit more jagged than the other ones.

In Fig. 7, an equivalent plot is given for the IceCube-Gen2 dataset. Again, all of the constrained GANs match the distributions very well.

### B.4 Runtime and parameter tuning

Table 7: Runtime (in hours) for one run of 100 000 iterations.

|  | WGAN | WGAN + pc (pcGAN) | Wu et al. | Wu et al. + pc | WGAN-GP | WGAN-GP + pc | SNGAN | SNGAN + pc |
|---|---|---|---|---|---|---|---|---|
| Synthetic (5.2) | 1.20 | 1.95 | 1.38 | 2.06 | 1.91 | 2.68 | 1.40 | 2.07 |
| IceCube-Gen2 (5.3) | 0.37 | 0.46 | 0.45 | 0.46 | 0.51 | 0.56 | 0.60 | 0.67 |

The runtime required for one run of 100 000 iterations for the various models and for the different datasets is given in Table 7. The time required to extract the representations $p_{\text{true}}$ for all of the constrained statistics combined is negligible in comparison: for the synthetic dataset, it takes 66 seconds, and for the IceCube-Gen2 dataset 0.05 seconds. These numbers have been obtained on a system with NVIDIA RTX 3060 Ti 8GB GPU, Intel Core i7 7700-K @ 4.2GHz CPU, and 16GB RAM. Overall, adding the probabilistic constraint increases the runtimes of the corresponding base GANs by around $40\% - 60\%$ in case of the synthetic dataset, where 101 statistics are matched. For the IceCube-Gen2 dataset, where only 2 statistics are matched, the increases in runtime are significantly lower at less than 30%.

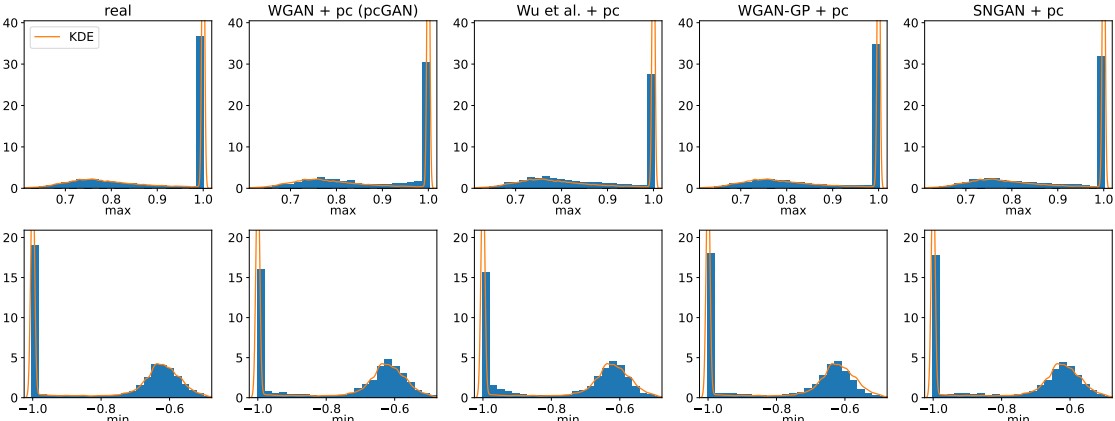

Figure 7: **(IceCube-Gen2)** The distributions of minimum and maximum values as obtained with the different GAN variants when combined with the probabilistic constraint are depicted, where the orange lines show the true distribution as obtained via KDE (7). For the histograms, $20\,000$ generated samples have been considered (or the full dataset, in case of the real distribution). Parameters: $bs = 256$, $\epsilon = 0.9$, $h=$KL, $\lambda = 2$.

Apart from the increased runtime for individual runs, new hyperparameters get introduced with the probabilistic constraint which will cause additional upfront cost when tuning the model. They include the global weighting coefficient $\lambda$, the parameter $\epsilon$ determining the amount of historical averaging, and the function $h$ to quantify the mismatch between the distributions. In principle, the heuristic choices in the formula for $\lambda_s$ can also be finetuned. The values used throughout the paper, e.g. those given in Tables 1 and 3, should serve as good starting points for these hyperparameters. The most important parameter to tune individually for each dataset is the global weighting coefficient $\lambda$ in (3). Assuming that the underlying unconstrained GAN has already been well-tuned, obtaining good results with the probabilistically constrained GAN should be possible with 5-10 additional runs.

The fine-tuning process of the pcGAN can add a sizable amount of time to the model development. In cases where the simulation would take minutes or hours to generate a single sample, and where thousands or millions of samples need to be generated, the pcGAN can still provide speedups. Apart from that, GANs have the advantage of being differentiable, which allows for their use in end-to-end optimization pipelines, e.g. for detector design (Dorigo et al., 2023). Hence, being faster than the traditional simulation is not always essential for GANs to be useful.

## C  Details on the Experiments

In this appendix, we give additional information on the experiments conducted in Sections 5.2-5.3, in particular on the network architectures and the training parameters. The code for the project is available at https://github.com/ppilar/pcGAN; note, however, that only the data for the synthetic example is available there.

### C.1  Constraints and High-level Features

We start by giving an overview of the different quantities that have been employed either as constraints or performance metrics.

For the 1D signals $x$ of length $N_x = 200$ in Sections 5.2 and 5.3, we used the minimum and maximum values, $\min = \min(x_0, \ldots, x_{N_x-1})$, $\max = \max(x_0, \ldots, x_{N_x-1})$, mean values, $\text{mean} = \frac{1}{N_x} \sum_{i=0}^{N_x-1} x_i$, the mean absolute values, $\text{mean(abs)} = \frac{1}{N_x} \sum_{i=0}^{N_x-1} |x_i|$, the number of zero crossings, $N_{\text{zc}}$, and the number of maxima, $N_{\text{max}}$, of the curves.

The discrete Fourier transform for real-valued inputs (as implemented in torch.fft.rfft) was utilized to obtain the complex Fourier coefficients for positive frequencies $k \in [0, \lfloor \frac{N_x}{2} \rfloor + 1]$ below the Nyquist frequency,

$$X_k = \sum_{n=0}^{N_x - 1} x_n e^{-i2\pi \frac{kn}{N_x}}, \tag{17}$$

and the corresponding power spectrum components are obtained as $S_k = \frac{1}{N_x}|X_k|^2$. The total spectral energy is then calculated as $S = \sum_{k=0}^{\lfloor \frac{N_x}{2} \rfloor + 1} S_k$. When employed as constraints, we did not constrain on the power spectrum components directly, but instead on $\mathrm{ps}[k] = \sqrt{N_x S_k}$.

For the different experiments, we considered the following set of high-level features: mean, mean(abs), max-min, E, $N_{zc}$, and $N_{\max}$. For the IceCube-Gen2 experiment, we omitted the mean, since it did not exhibit interesting structure in its distribution.

## C.2 Synthetic Example (Section 5.2)

The synthetic dataset consists of $100\,000$ samples of size 200, generated as described in Section 5.2. For this example, we employed convolutional networks for both the discriminator and generator; details on the corresponding network architectures are given in Tables 9 and 10, respectively. In layers where both batch normalization and an activation function are listed in the column 'Activation', batch normalization is applied to the input of the layer whereas the activation function is applied to the output. Padding is employed in each layer such that the given output sizes are obtained; reflection padding is utilized.

Table 8: Hyperparameters used for the experiments.

| Experiment | $N_{\mathrm{avg}}$ | $N_{\mathrm{it}}$ | lr | $f_{\mathrm{sched}}$ | $\mathrm{it}_{\mathrm{sched}}$ | $\beta_1$ | $\beta_2$ | clamping |
|---|---|---|---|---|---|---|---|---|
| Synthetic (5.2) | 50 | $100\,000$ | 2e-4 | 0.5 | 70000 | 0 | 0.9 | $[0, 0.005]$ |
| IceCube-Gen2 (5.3) | 50 | $100\,000$ | 5e-4 | 0.5 | 40000 | 0 | 0.9 | $[0, 0.1]$ |

In Figure 8, the search for the best values $f_\sigma^s$ in (8) is illustrated for $h = \mathrm{KL}$. It is apparent, that there is a clear, batch size-dependent minimum of the KL-divergence for each constraint, with larger batch sizes tending towards larger values of $f_\sigma^s$; this is due to the fact that more samples in the minibatch allow for a more fine-grained approximation of the generated distribution. In the top right plot, optimal values of $f_\sigma^s$ are depicted for all components $\mathrm{ps}[i]$ of the power spectrum. The spike around $i \approx 10$ is the result of some outliers in the values of the power spectrum components; they lead to a high standard deviation of the true distribution, which in turn requires a large $f_\sigma^s$ in order to obtain small enough standard deviations for the KDE to resolve the narrow peak well.

In the bottom row, approximations of the generated distributions as obtained via the minibatches are depicted. It is apparent that the mixtures of Gaussians approximate them reasonably well, with larger batch sizes typically giving better results.

In Figure 10, samples from the true distribution as well as generated samples from the different GANs are depicted. All of the GANs produce reasonable-looking results, although upon closer inspection it becomes apparent that they do not necessarily constitute a superposition of two sine waves. Only the WGAN seems to have a tendency to produce rugged curves.

## C.3 IceCube-Gen2 (Section 5.3)

For this example, we considered $50\,000$ IceCube-Gen2 radio-detector signals of size 200 (generated using the NuRadioMC code (Glaser et al., 2020) according to the ARZ algorithm (Alvarez-Muñiz et al., 2010)), normalized to their respective maximum absolute values. The networks employed are a mixture of convolutional and fully connected networks, which have been based on the architectures used in Holmberg (2022); details on discriminator and generator architectures are given in Tables 11 and 12, respectively. For the discriminator, the input to the network is first fed through four convolutional layers in parallel, the outputs of which are

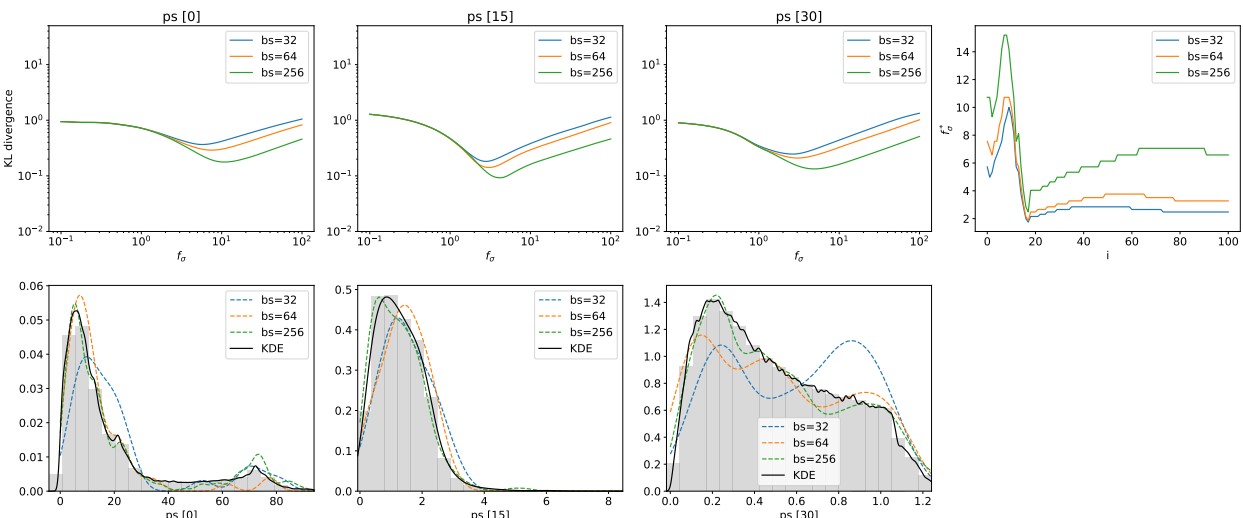

Figure 8: **(Synthetic example)** Determining optimal values $f_\sigma^s$ for $h = \mathrm{KL}$. **Top left:** The three plots on the top left depict the dependency of the KL divergence on the factor $f_\sigma$ for different power spectrum components; the curves have been averaged over 50 minibatches sampled from the original dataset. **Bottom left:** The first three plots in the bottom row depict the distribution of the constraint values together with their KDE representation, as well as curves obtained via (8) from minibatches of different size (not averaged); it is between them that the KL divergences in the top row have been calculated. **Top right:** Optimal values of the factor $f_\sigma^s$ are depicted for different batch sizes, where the index $i$ gives the respective component of the power spectrum.

subsequently concatenated into one long array. The LeakyReLU activation function with factor 0.2 is applied. During training, we also check for a rare failure mode where the GAN generates only horizontal lines; if this happens, the training is restarted.

In Figure 9, a plot on the process of determining optimal values for $f_\sigma^s$ is shown at the example $h = \mathrm{KL}$. Same as for the synthetic example (compare Fig. 8), the KL divergences as a function of $f_\sigma$ exhibit clear minima that depend on the batch size.

In Figure 12, samples from the true distribution as well as generated samples from the different GANs are depicted. Altogether, most of the generated samples look good, with none of the models clearly outperforming the others.

## C.4  Training Parameters

Here, we summarize the training parameters used for the different experiments. $N_{\mathrm{it}}$ gives the number of training iterations, lr the learning rate, and $\lambda$ the weighting factor for the constraints in (3). In the column 'clamping', the range is given to which network parameters of the discriminator $D$ were clamped in order to enforce the Lipschitz constraint in WGANs (Arjovsky et al., 2017). The ADAM optimizer (Kingma and Ba, 2015) was used for training the networks, with hyperparameter values $\beta_1$ and $\beta_2$; a scheduler was employed that reduced the learning rate by a factor of $f_{\mathrm{sched}}$ after $it_{\mathrm{sched}}$ iterations. The weighting factor for the statistical constraint from Wu et al. (2020) was chosen as $\lambda_{\mathrm{Wu}} = 1$. The weighting factor for the gradient penalty in WGAN-GP was chosen as $\lambda_{\mathrm{GP}} = 10$. The parameter $m$, which gives the number of discriminator updates per generator update, was chosen as 1.

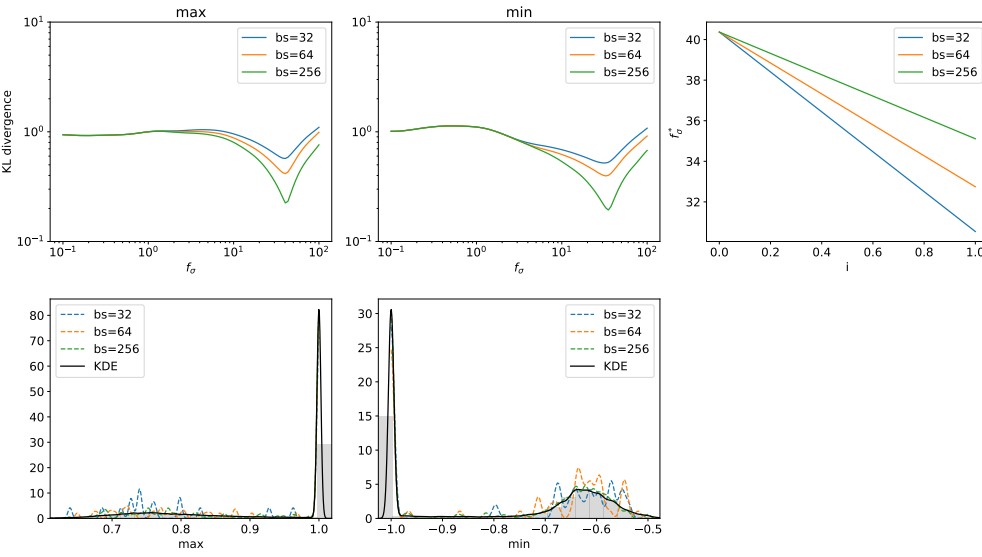

Figure 9: **(IceCube-Gen2)** Determining optimal values $f_\sigma^s$ for $h = \mathrm{KL}$. **Top left:** The two plots on the top left depict the dependency of the KL divergence on the factor $f_\sigma$; the curves have been averaged over 50 minibatches sampled from the original dataset. **Bottom left:** The first two plots in the bottom row depict the distributions of the constraint values together with their KDE representation, as well as curves obtained via (8) from minibatches of different size (not averaged); it is between them that the KL divergences in the top row have been calculated. **Top right:** Optimal values of the factor $f_\sigma^s$ are depicted for different batch sizes, where the index $i$ gives the respective constraint.

Table 9: Discriminator architecture for the synthetic example.

| Layer | Output Size | Kernel Size | Stride | Activation |
|---|---|---|---|---|
| Input | $1 \times 200$ | | | |
| Conv | $32 \times 99$ | 3 | 2 | BatchNorm, ReLU |
| Conv | $32 \times 99$ | 3 | 1 | BatchNorm, ReLU |
| Conv | $32 \times 99$ | 3 | 1 | ReLU |
| Conv | $64 \times 48$ | 3 | 2 | BatchNorm, ReLU |
| Conv | $64 \times 48$ | 3 | 1 | BatchNorm, ReLU |
| Conv | $64 \times 48$ | 3 | 1 | ReLU |
| Conv | $128 \times 23$ | 3 | 2 | ReLU |
| Conv | $128 \times 23$ | 3 | 1 | ReLU |
| Conv | $128 \times 23$ | 3 | 1 | ReLU |
| Conv | $256 \times 10$ | 3 | 2 | ReLU |
| Conv | $256 \times 10$ | 3 | 1 | ReLU |
| Conv | $256 \times 10$ | 3 | 1 | ReLU |
| Flatten | 2560 | | | |
| Linear | 1 | | | |

Table 10: Generator architecture for the synthetic example.

| Layer | Output Size | Kernel Size | Stride | Activation |
|---|---|---|---|---|
| Input | $1 \times 5$ | | | BatchNorm |
| ConvTransp | $256 \times 25$ | 3 | 16 | BatchNorm, Tanh |
| Conv | $256 \times 25$ | 3 | 1 | BatchNorm, Tanh |
| Conv | $256 \times 25$ | 3 | 1 | BatchNorm, Tanh |
| ConvTransp | $128 \times 50$ | 3 | 2 | BatchNorm, Tanh |
| Conv | $128 \times 50$ | 3 | 1 | BatchNorm, Tanh |
| Conv | $128 \times 50$ | 3 | 1 | BatchNorm, Tanh |
| ConvTransp | $64 \times 100$ | 3 | 2 | BatchNorm, Tanh |
| Conv | $64 \times 100$ | 3 | 1 | BatchNorm, Tanh |
| Conv | $64 \times 100$ | 3 | 1 | BatchNorm, Tanh |
| ConvTransp | $32 \times 200$ | 3 | 2 | BatchNorm, Tanh |
| Conv | $32 \times 200$ | 3 | 1 | BatchNorm, Tanh |
| Conv | $32 \times 200$ | 3 | 1 | Tanh |
| Conv | $1 \times 200$ | 3 | 1 | Tanh |

Table 11: Discriminator architecture for the IceCube-Gen2 data. The input is first fed through the layers Conv01-Conv04 in parallel and the outputs are subsequently concatenated into one long array.

| Layer | Output Shape | Kernel Size | Stride | Activation |
|---|---|---|---|---|
| Input | $1 \times 200$ | | | |
| Conv01 | $32 \times 49$ | 5 | 4 | LeakyReLU |
| Conv02 | $32 \times 47$ | 15 | 4 | LeakyReLU |
| Conv03 | $32 \times 44$ | 25 | 4 | LeakyReLU |
| Conv04 | $32 \times 42$ | 35 | 4 | LeakyReLU |
| Concatenate | $32 \times 182$ | | | |
| Conv | $1 \times 182$ | 1 | 1 | LeakyReLU |
| Linear | 92 | | | LeakyReLU |
| Linear | 45 | | | LeakyReLU |
| Linear | 20 | | | LeakyReLU |
| Linear | 1 | | | |

Table 12: Generator architecture for the IceCube-Gen2 data.

| Layer | Output Size | Kernel Size | Stride | Activation |
|---|---|---|---|---|
| Input | 5 | | | |
| Linear | 24 | | | ReLU |
| Conv | $48 \times 24$ | 3 | 1 | ReLU |
| Conv | $48 \times 24$ | 3 | 1 | ReLU |
| ConvTransp | $24 \times 49$ | 3 | 2 | ReLU |
| Conv | $24 \times 49$ | 3 | 1 | ReLU |
| ConvTransp | $12 \times 99$ | 3 | 2 | ReLU |
| Conv | $12 \times 99$ | 3 | 1 | ReLU |
| ConvTransp | $6 \times 199$ | 3 | 2 | ReLU |
| Conv | $1 \times 200$ | 4 | 1 | |

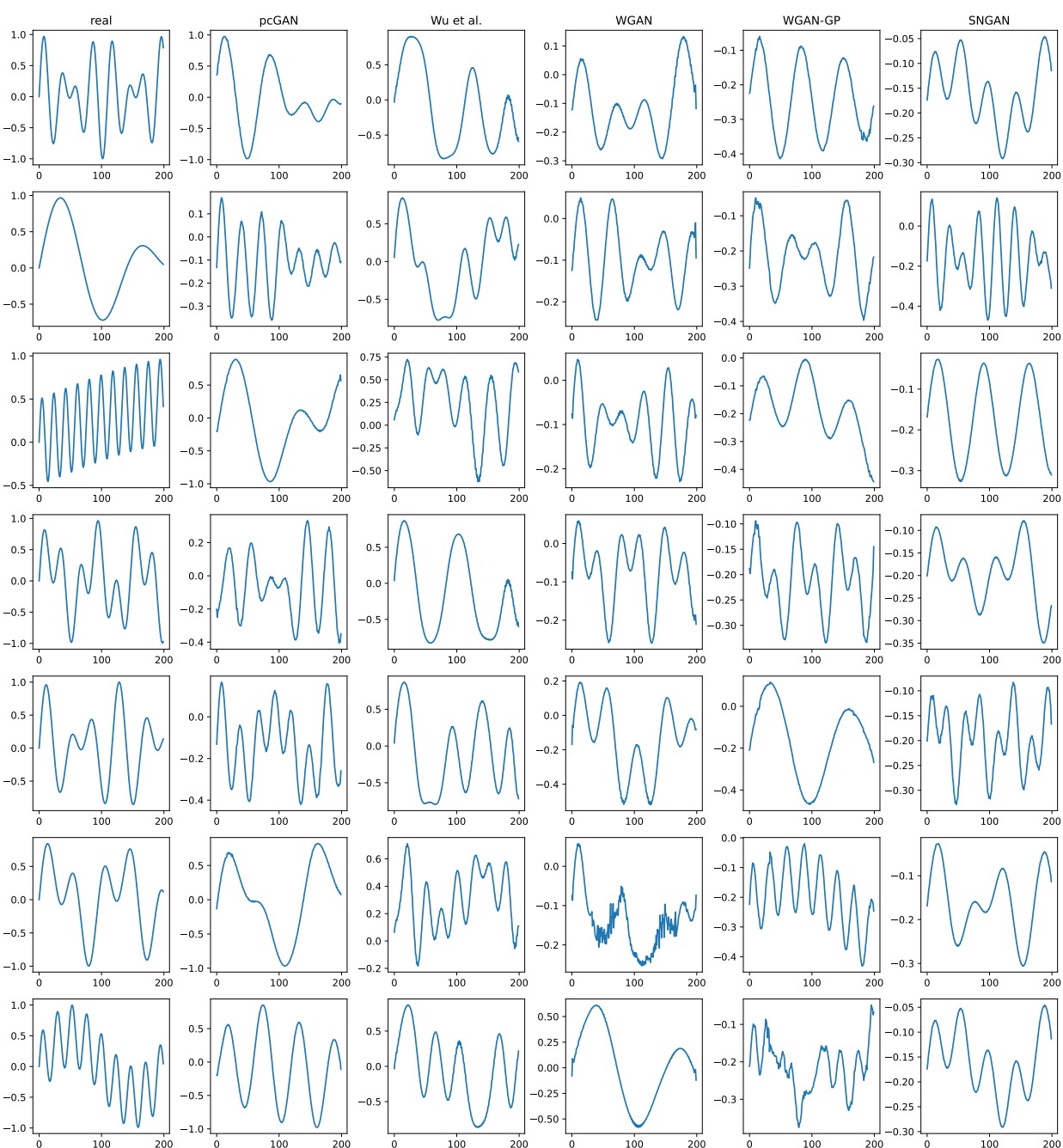

Figure 10: Samples for the synthetic example as obtained from the different models.

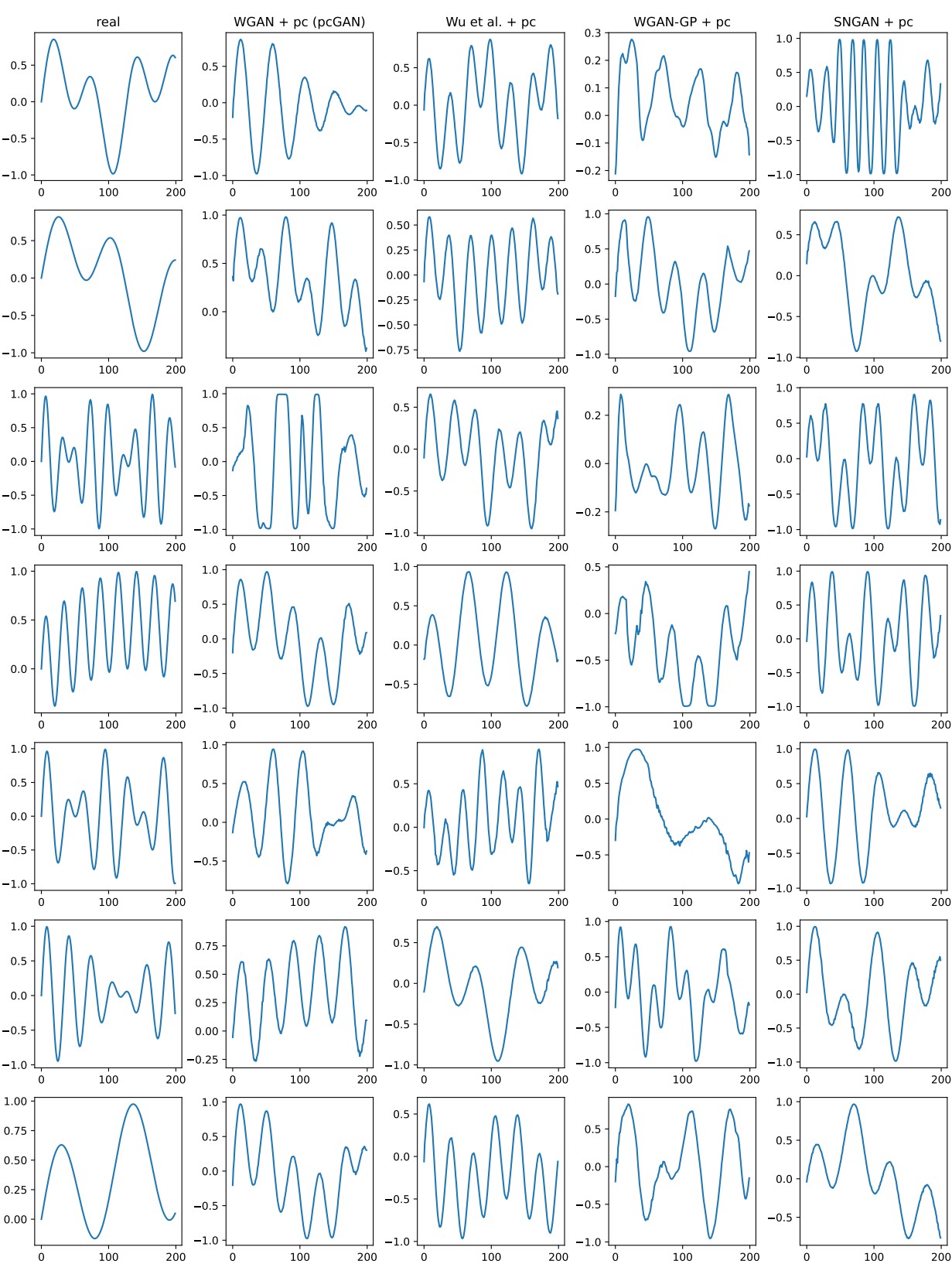

Figure 11: Samples for the synthetic example as obtained from the different constrained models.

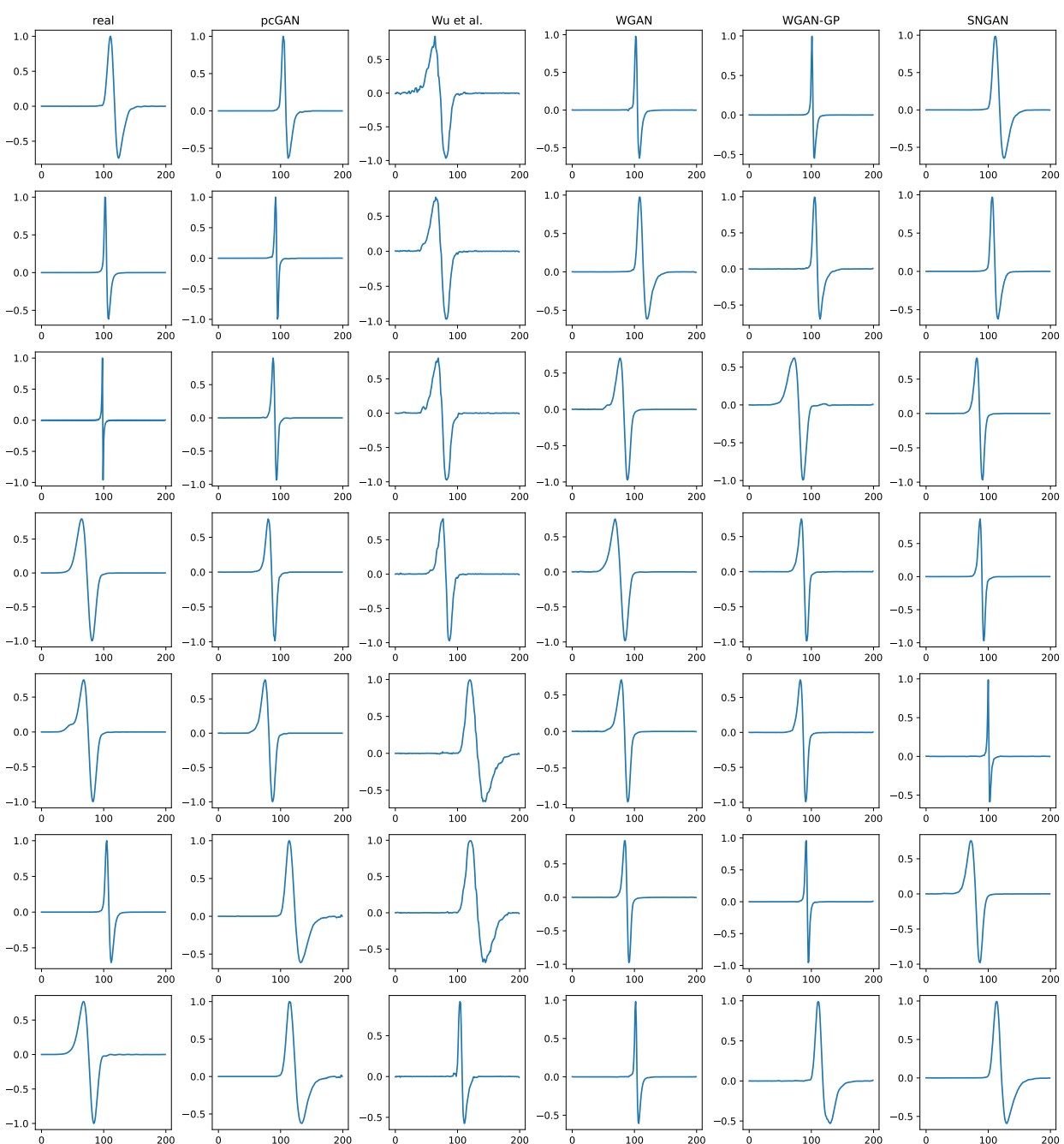

Figure 12: Samples for the IceCube-Gen2 dataset as obtained from the different models.

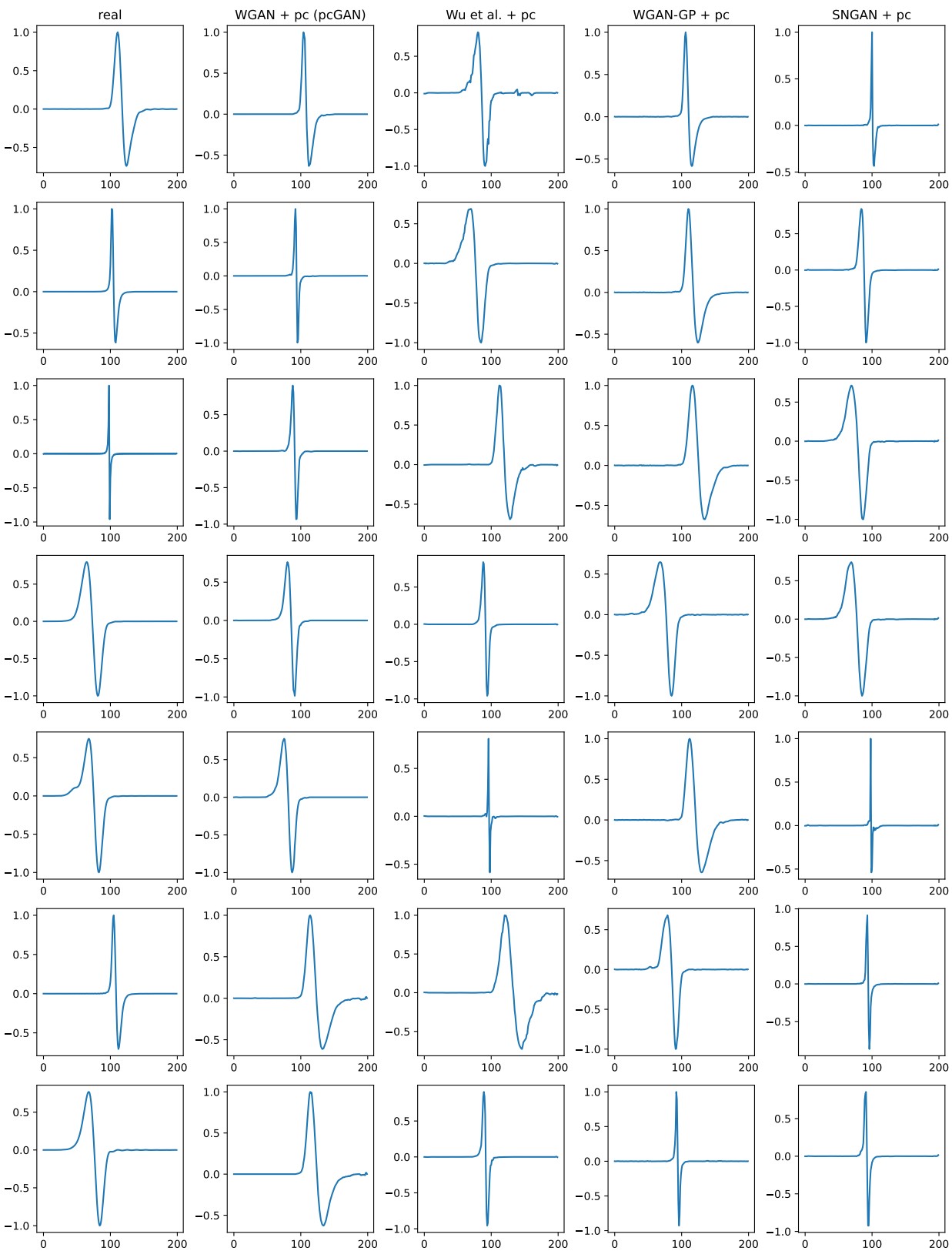

Figure 13: Samples for the IceCube-Gen2 dataset as obtained from the different constrained models.

