# OpenReview forum: "Probabilistic Matching of Real and Generated Data Statistics in Generative Adversarial Networks"
_TMLR — Accepted by TMLR_

### Review · Reviewer_PNEX · 2024-07-16

**Summary Of Contributions:**

In the context of data generation for scientific applications, the authors introduce a method to align the distributions of statistics of generated data with those of real data. Their approach is suited for Generative Adversarial Networks (GANs) and enhances the generator's loss function by incorporating a new term that measures distributional differences using f-divergences. By applying kernel density estimation, they represent the real distributions and estimate the corresponding generated distributions at each iteration. This method referred to as probabilistically constrained GAN (pcGAN), focuses on matching the overall shapes of distributions rather than individual sample properties. The approach is particularly beneficial for scientific applications, as domain-specific differentiable statistics can be selected. The authors perform the empirical evaluation on one synthetic and one real-world dataset.

**Audience:**

Yes

**Claims And Evidence:**

No

**Requested Changes:**

### 1. Empirical Evaluation (Critical)

- Increase the number of runs per model to strengthen the results.
- Evaluate the addition of the probabilistic constraints to different GAN variants on real data
- In Section 5.2 (pag. 9) authors claim "In therm of execution time, the pcGAN takes about twice as long to train as the unconstrained WGAN". This claim needs to be supported with numerical results.
- Adding the new loss requires adjusting several parameters ($f_\sigma$, $\lambda$, $\epsilon$, batch size). How expensive is the fine-tuning process?

### 2. Results Presentation (Critical)

The graphical representation chosen by the authors to report the numerical results is 1) not easy to read and 2) not compact, forcing the authors to move important results in the Appendix. For example, the authors describe four metrics used in the empirical evaluation, but in Figures 4 and 5 they report only two of them, without providing any explanation for the missing results.
Thus, I strongly suggest the authors to:
  - replace Figures 3, 5, and 7 with tables, and include the results for all the metrics;
  - replace Figure 4 with a line chart with a confidence interval, and include the results for all the  metrics.

**Strengths And Weaknesses:**

## Strengths

- The paper introduces a further form of supervision in training GANs for scientific applications to enforce a better match between generated and real data statistics.
- The proposed idea is presented with clarity, and the author's contribution is well-stated.
- The authors provide the source code to reproduce their results.
- The paper is well-written and well-organized.

## Weaknesses


- The empirical evaluation is not convincing (see Requested Changes - 1):

  - The number of runs (3 per model) is not enough to produce a convincing empirical evaluation, as the differences in performance cannot be proven to be statistically relevant with only 3 evaluations per model.

  - The experiments conducted on the real dataset are not exhaustive since they do not include the evaluation of adding the probabilistic constraints to different GAN variants, as done for the synthetic dataset.

  - The numerical results reported in Figure 7 do not show a clear improvement in pcGAN compared to the other models.

  - The new loss term in pcGAN introduces an overhead both in hyper-parameters fine-tuning and training procedure. However, this aspect is not covered in the paper. As pointed out by the authors in the introduction, "GANs may serve as surrogate models for expensive but highly accurate numerical simulations", thus the tradeoff between performance improvement and computational overhead is crucial for their method, but it is missing in the analysis.

- The results are not well-presented (see Requested Changes - 2).

---

> ### Author Response · Authors · 2024-07-19
>
> We thank the reviewer for the careful evaluation of our paper. We will incorporate the requested changes in the revised version. In the meantime, we would like to point out that the source code for the project is available in the supplementary material. In the readme file, a link to download the real world dataset is also included. We apologize for forgetting to mention this in the paper.

---

> > ### Comment · Reviewer_PNEX · 2024-07-19
> >
> > Thank you for your clarification. I checked the code, and it is complete and working. I've updated my review accordingly.

---

> ### Author Response · Authors · 2024-08-06
>
> Thank you again for the constructive feedback.
>
> To comment on the weaknesses and requested changes:
>
> > The empirical evaluation is not convincing (see Requested Changes - 1):
> > The number of runs (3 per model) is not enough to produce a convincing empirical evaluation, as the differences in performance cannot be proven to be statistically relevant with only 3 evaluations per model.
>
> To improve the empirical evaluation, we have increased the number of runs to 10 per model.
>
> > The experiments conducted on the real dataset are not exhaustive since they do not include the evaluation of adding the probabilistic constraints to different GAN variants, as done for the synthetic dataset.
>
> In the revised version, we conduct additional experiments on the real dataset where we added the probabilistic constraint to the different GAN variants (see Table 3).
>
> > The numerical results reported in Figure 7 do not show a clear improvement in pcGAN compared to the other models.
>
> We now report the results in Table 3. Adding the probabilistic constraint improves most of the GAN variants and is effective at matching the constraints. It is true that the standard pcGAN is not better than the other GAN variants in this example. To account for this, we rephrased the second paragraph in the conclusions section.
>
> > The new loss term in pcGAN introduces an overhead both in hyper-parameters fine-tuning and training procedure. However, this aspect is not covered in the paper. As pointed out by the authors in the introduction, "GANs may serve as surrogate models for expensive but highly accurate numerical simulations", thus the tradeoff between performance improvement and computational overhead is crucial for their method, but it is missing in the analysis.
>
> We now discuss this matter in Appendix B.4. Even in cases where the GAN runtime may exceed that of the traditional simulation, there can still be use cases for the GAN due to its differentiability, e.g. in detector design.
>
> > The results are not well-presented (see Requested Changes - 2).
>
> Thank you for the suggested improvements. We have updated the presentation of the results accordingly.

---

> > ### Comment · Reviewer_PNEX · 2024-08-08
> >
> > I thank the authors for their reply and their effort in addressing all the requested changes.
> > The revised version of the paper provides convincing and clear evidence to support the authors' claims.

---

### Review · Reviewer_uLxz · 2024-07-26

**Summary Of Contributions:**

This paper proposes the probabilistically constrained GAN to improve the statistical fidelity of generated samples in GANs. The main contribution is a technique to match the distributions of certain statistics between real and generated data by adding a new loss term to the generator loss function. This approach aims to ensure that generated samples not only look realistic but also accurately reflect the true data distribution.

**Audience:**

Yes

**Broader Impact Concerns:**

I do not see any significant broader impact concerns that would necessitate adding a Broader Impact Statement or expanding an existing one.

**Claims And Evidence:**

Yes

**Requested Changes:**

An intriguing aspect to consider is the potential adaptability of the proposed method to other probabilistic generative models, such as energy-based models (EBMs). It would be valuable to explore how the core principles of the pcGAN approach might be applied beyond the GAN framework. Could the authors comment on the feasibility and potential benefits of extending their proposed model to these broader classes of generative models?

**Strengths And Weaknesses:**

Strengths:

1. This paper introduces a new way to constrain GANs by matching distributions of chosen statistics.

2. The use of f-divergences and kernel density estimation is well-justified.

3. The approach can be applied to various GAN architectures and allows for arbitrary differentiable statistics to be constrained.

4. The experiments cover both synthetic and real-world datasets, with a wide range of evaluation metrics and comparisons to existing methods. This paper also includes the detailed ablation studies.

Weaknesses:

1. The use of kernel density estimation may become computationally expensive for high-dimensional statistics. A more detailed discussion on the method's scalability would be beneficial. Additionally, the paper could be strengthened by demonstrating improvements on existing state-of-the-art GAN models for more complex tasks like image generation, rather than focusing primarily on toy datasets.

2. While the paper makes good use of KDE to approximate statistics, it relies on hand-crafted features. An interesting extension would be to explore the use of learned statistics from existing models, which could potentially capture more complex or nuanced aspects of the data distribution. This approach could draw connections to the rich history of analysis-by-synthesis methods in statistical modeling.

---

> ### Author Response · Authors · 2024-08-06
>
> We thank the reviewer for the careful evaluation of our paper.
>
> To address the weaknesses:
>
> > The use of kernel density estimation may become computationally expensive for high-dimensional statistics. A more detailed discussion on the method's scalability would be beneficial.
>
> In this paper, we do not take into account correlations between statistics and therefore end up with one-dimensional distributions to match; for high-dimensional statistics, the method may need to be adapted. We point this out in the future work section. It may be possible to use energy-based models to represent the distributions instead of KDE in such cases.
>
> > While the paper makes good use of KDE to approximate statistics, it relies on hand-crafted features. An interesting extension would be to explore the use of learned statistics from existing models, which could potentially capture more complex or nuanced aspects of the data distribution.
>
> This is an interesting suggestion and we have added a comment in the future work section.
>
> To address the requested changes:
>
> > An intriguing aspect to consider is the potential adaptability of the proposed method to other probabilistic generative models, such as energy-based models (EBMs). It would be valuable to explore how the core principles of the pcGAN approach might be applied beyond the GAN framework. Could the authors comment on the feasibility and potential benefits of extending their proposed model to these broader classes of generative models?
>
> The probabilistic constraint should also work with other generative models, where a suitable spot to put the additional loss term can be identified. The main ingredient for this to be possible would be that new samples are generated during each training iteration. Now we also emphasize this in the future work section.

---

### Review · Reviewer_1V6h · 2024-07-29

**Summary Of Contributions:**

This paper proposes a novel regularization mechanism for GANs based on divergences between some differentiable statistics of the true and generated distributions. In particular in this work, those statistics are chosen based on domain knowledge and estimated with KDEs. Some additional work is done in terms of "tricks" that are part of them method, such as how to chose kernel parameters, or the weighing of divergences.
Empirically, the method is tested on time series data where it can be observed that the proposed method better recovers the desired statistics. A number of design choices are also tested.

**Audience:**

Yes

**Broader Impact Concerns:**

No concerns.

**Claims And Evidence:**

Yes

**Requested Changes:**

The main change I'd like to see in this paper is better justification for some design choices, that appear to be tricks set by trial and error rather than methodically. For example, why set $\bar\sigma_s$ the way it is set? This is just a hyperparameter that should be empirically tuned. Similarly, the method to determine $f_\sigma^s$ appears very heuristic, with some constants sprinkled in Algorithm 3. The method to determine $\lambda_s$ is just as suspicious; where does this $0.1$ value (or even really the whole $\eta$ formula) come from?

Some text comments:
- "A major advantage of using f-divergence..." this is fairly constrained for situations where one can easily integrate over x. This is rarely the case for a complex enough distribution (e.g. one parameterized by a deep model).
- Equation 5 is wrong and shows the Jeffreys divergence rather than the JS divergence. The two are quite close but the difference matters numerically and empirically (what are the experiments using?)
- "We chose to employ KDE with Gaussian kernels", this was a bit scary to read, but I was reassured later on by appendix B.1. I'd recommend presenting the method in a manner that's a bit more generic (but using Gaussian kernels as a running example may be helpful for readers).
- Equation 10 suggests some recursive computation, is this really the case? If so is it truncated?
- Writing `linspace(0, 20, 200)` belongs more in a technical report than a scientific paper. Consider revising.

**Strengths And Weaknesses:**

Overall I appreciate what the method is pushing for, and it seems like an elegant way of incorporating prior knowledge. The paper is clear and easy to read, I think I could almost reimplement this just from the text.

While the empirical evaluation is limited, there's still some good work done there where the synthetic example is a good illustration, and some design choices are tested. Unfortunately, there are still some choices that seem not very well justified (see below). It also feels like there's a missed opportunity here to test on way more domains with different priors. This could be a much more general paper considering how many domains this method could apply to, and this lack of scope is perhaps the main weakness of the paper.

On the novelty side, this is novel to me but I'm not up-to-date enough on this corner of the GAN world to be an authority here.

---

> ### Author Response · Authors · 2024-08-06
>
> We thank the reviewer for the careful evaluation of our paper.
>
> > The main change I‘d like to see in this paper is better justification for some design choices, that appear to be tricks set by trial and error rather than methodically. For example, why set $\bar \sigma_s$ the way it is set? This is just a hyperparameter that should be empirically tuned.
>
> The main purpose of $\bar \sigma_s$ is to represent the true distributions well. The leftmost plots in Figs. 1, 2 and 5 show that this is the case and therefore we do not think that further tuning is required here. We now point this out in the paper.
>
> > Similarly, the method to determine $f_{\sigma}^s$ appears very heuristic, with some constants sprinkled in Algorithm 3.
>
> The algorithm to determine $f_{\sigma}^s$ is motivated by the thought that KDE approximations obtained from minibatches sampled from true data should fit the true data distribution as well as possible. We now make this motivation more explicit in Appendix A. In Algorithm 3, we chose to average over $N_{\rm avg}=50$ minibatches to determine the values $f_{\sigma}^s$. We now list the value for $N_{\rm avg}$ in Table 8 instead, together with other parameters. Introducing the standard deviation of the constraint values via $c$ into the definition of $\sigma$ has the effect that the optimal value $f_{\sigma}^s$ will be more similar for different statistics, which is advantageous for the grid search. The logspace that we defined is the grid over which we search for the optimal value. Figures 8 and 9 show that the minima are indeed within this range.
>
> > The method to determine $\lambda_s$ is just as suspicious; where does this 0.1 value (or even really the whole $\eta$ formula) come from?
>
> It is correct that the relative weighting of the different statistics via $\eta$ is heuristic (but has proven to work well). The first term in the definition of $\eta$ quantifies the constraint fulfillment relative to the best-fulfilled one and the second term prevents already well-matched statistics from ceasing to be included in the constraint. We now make this clear in the text.
>
> About the text comments:
>
> > "A major advantage of using f-divergence..." this is fairly constrained for situations where one can easily integrate over x.
>
> In our work, we do not consider correlations between the statistics and therefore end up with one-dimensional integrals, for which the integration is feasible. We allude to higher-dimensional statistics in the future work section.
>
> > Equation 5 is wrong and shows the Jeffreys divergence rather than the JS divergence.
>
> Thank you for pointing this out, it is indeed Jeffreys divergence that was used. We apologize for the mistake and have corrected it in the new version of the paper.
>
> >  I'd recommend presenting the method in a manner that's a bit more generic (but using Gaussian kernels as a running example may be helpful for readers).
>
> Thank you for the recommendation. We have adapted the formulation in the revised version (see Section 4.2 and Appendices B.1 and B.2).
>
> > Equation 10 suggests some recursive computation, is this really the case? If so is it truncated?
>
> This equation only requires the result from the previous iteration, which can be kept in memory and therefore no recursion is required.
>
> > Writing linspace(0, 20, 200) belongs more in a technical report than a scientific paper. Consider revising.
>
> We revised the formulation.

---

### Decision · Action_Editor_cFhw · 2024-08-28

**Recommendation:** Accept as is

**Comment:**

The paper does not qualify for a certification.

Overall, the paper meets the acceptance criteria of TMLR. The claims are supported by convincing experiments, and the proposed solution and analysis could be of interest to those working in the area of generative modelling and their applications to time series data.

**Audience:**

The work can spark some interest in the community of generative modelling, specifically on trade-offs between quality and fidelity of generated data, as well as on the incorporation of prior knowledge.

**Claims And Evidence:**

The paper addresses an important problem in generative modeling: how to generate high-quality data while ensuring high statistical fidelity to real-world data. This is achieved by proposing an objective for training a generative adversarial network and testing it on time series data. This objective introduces a new regularization term in the generator loss, which ensures that the distribution statistics of real and generated data are the same. Specifically, a set of manually extracted feature statistics (e.g., mean, absolute mean, number of zero crossings, etc.) is first extracted from both real and generated data. Kernel density estimation is then applied to obtain the corresponding distributions, and finally, the regularizer is computed using the f-divergence between these two distributions.

Initially, reviewers pointed out some issues related to the quality of the presentation and the experimental methodology. However, the authors effectively addressed these concerns during the discussion phase by providing (i) an explanation of design choices and hyperparameters, and (ii) additional experiments/explanations to underscore the significance of the results.

A notable weakness of the current work relates to the scope of the experimental analysis. Given the generality of the proposed method, the reviewers expected a broader analysis with higher-dimensional data and/or a demonstration that goes beyond manually extracted feature statistics. While these requests were not directly addressed, they were discussed in the paper as limitations and therefore deferred to future work.

Overall, the paper meets the acceptance criteria of TMLR. The claims are supported by convincing experiments, and the proposed solution and analysis could be of interest to those working in the area of generative modelling and their applications to time series data. Therefore, I recommend its acceptance.